# Structural insights into actin isoforms

Amandeep S Arora[1], Hsiang-Ling Huang[1], Ramanpreet Singh[1], Yoshie Narui[2], Andrejus Suchenko[3], Tomoyuki Hatano[3,4], Sarah M Heissler[1], Mohan K Balasubramanian[3,4], Krishna Chinthalapudi[1]*

[1]Department of Physiology and Cell Biology, Dorothy M. Davis Heart and Lung Research Institute, The Ohio State University College of Medicine, Columbus, United States; [2]Center for Electron Microscopy and Analysis, The Ohio State University, Columbus, United States; [3]Centre for Mechanochemical Cell Biology and Warwick Medical School, Division of Biomedical Sciences, Coventry, United Kingdom; [4]University of Warwick, Coventry, United Kingdom

**Abstract** Actin isoforms organize into distinct networks that are essential for the normal function of eukaryotic cells. Despite a high level of sequence and structure conservation, subtle differences in their design principles determine the interaction with myosin motors and actin-binding proteins. Therefore, identifying how the structure of actin isoforms relates to function is important for our understanding of normal cytoskeletal physiology. Here, we report the high-resolution structures of filamentous skeletal muscle α-actin (3.37 Å), cardiac muscle α-actin (3.07 Å), ß-actin (2.99 Å), and γ-actin (3.38 Å) in the $Mg^{2+}$·ADP state with their native post-translational modifications. The structures revealed isoform-specific conformations of the N-terminus that shift closer to the filament surface upon myosin binding, thereby establishing isoform-specific interfaces. Collectively, the structures of single-isotype, post-translationally modified bare skeletal muscle α-actin, cardiac muscle α-actin, ß-actin, and γ-actin reveal general principles, similarities, and differences between isoforms. They complement the repertoire of known actin structures and allow for a comprehensive understanding of in vitro and in vivo functions of actin isoforms.

## Editor's evaluation

This study presents four high quality cryo-EM structures of ADP-actin filaments formed from skeletal α-, cardiac α-, cytoplasmic β- and cytoplasmic γ-actin. These structures are important for understanding the functional differences among these actin isoforms. This work is of significant general interest, because actin filaments, composed of different actin isoforms, have a critical role in a number of physiological processes from muscle contraction to cell migration and division.

## Introduction

Actin isoforms are among the most ubiquitous and abundant structural proteins that facilitate the functional organization of the cytoplasm of eukaryotic cells (*Blanchoin et al., 2014*; *Pollard, 2016*). Humans express six actin genes in a tissue-specific and developmentally regulated manner (*Kashina, 2020*). The gene products are structurally and functionally highly conserved among vertebrates and can be grouped into four muscle actins: skeletal muscle α-actin, smooth muscle α-actin (vascular), cardiac muscle α-actin, smooth muscle γ-actin (enteric), and two nonmuscle actins (ß-actin and γ-actin; *Otey et al., 1987*; *Vandekerckhove and Weber, 1978*). Most cells maintain a defined ratio of actin isoforms with muscle and nonmuscle actins representing the main isoforms in muscle and nonmuscle cells, respectively (*Kashina, 2020*; *Otey et al., 1987*; *Vandekerckhove and Weber, 1978*; *Kee et al., 2009*; *Tondeleir et al., 2009*; *Patrinostro et al., 2017*). Actin isoforms have specific and redundant

*For correspondence:
krishna.chinthalapudi@osumc.edu

**eLife digest** The protein actin is important for many fundamental processes in biology, from contracting muscle to dividing a cell in two. As actin is involved in such a variety of roles, human cells have slightly different versions of the protein, known as isoforms. For example, alpha-actin is vital for contracting muscle, while beta- and gamma-actin drive cellular processes in non-muscle cells.

In order to carry out its various functions, actin interacts with many other proteins inside the cell, such as myosin motors which power muscle contraction. These interactions rely on the precise chain of building blocks, known as amino acids, that make up the actin isoforms; even subtle alterations in this sequence can influence the behavior of the protein. However, it is not clear how differences in the amino acid sequence of the actin isoforms impact actin's interactions with other proteins.

Arora et al. addressed this by studying the structure of four human actin isoforms using a technique called cryo-electron microscopy, where the proteins are flash-frozen and bombarded with electrons. These experiments showed where differences between the amino acid chains of each isoform were located in the protein. Arora et al. then compared their structures with previous work showing the structure of actin bound to myosin. This revealed that the tail-end of the protein (known as the N-terminus) differed in shape between the four isoforms, and this variation may influence how actin binds to others proteins in the cell.

These results are an important foundation for further work on actin and how it interacts with other proteins. The structures could help researchers design new tools that can be used to target specific isoforms of actin in different types of laboratory experiments.

roles in cells and display different biochemistries, cellular localization, and interactions with myosin motors and actin-binding proteins (ABPs; *Pollard, 2016*; *Kashina, 2020*; *Perrin and Ervasti, 2010*; *Vedula et al., 2021*; *Varland et al., 2019*; *Bunnell et al., 2011*; *Tondeleir et al., 2012*; *Baranwal et al., 2012*; *Diensthuber et al., 2011*; *Müller et al., 2013*; *Lee and Dominguez, 2010*; *Harris et al., 2020*; *Lappalainen, 2016*). Driven by the dominating action of ABPs and signaling proteins, differences between actin isoforms may facilitate the formation of diverse cellular actin networks with distinct compositions, architectures, dynamics, and mechanics that enable fundamental cell functions including adhesion, migration, and contractility (*Blanchoin et al., 2014*; *Tondeleir et al., 2009*; *Vedula et al., 2021*). The altered expression and mutation of the genes encoding for actin isoforms have been linked to human diseases (*Tondeleir et al., 2009*; *Chaponnier and Gabbiani, 2004*; *Parker et al., 2020*).

Actin isoforms share high sequence identity at the protein level (~93–99%) and the propensity to self-assemble into helical, polarized filaments (F-actin) from monomers (G-actin; *Pollard, 2016*; *Perrin and Ervasti, 2010*; *Dominguez and Holmes, 2011*; *Arnesen et al., 2018*). Since the publication of the first crystal structure of G-actin in complex with DNaseI ~30 years ago, extensive studies have advanced our understanding of the structure of monomeric and filamentous actin, polymerization mechanisms, post-translational modifications (PTMs), interaction with drugs, myosin motors, and ABPs at ever-increasing resolution (*Dominguez and Holmes, 2011*; *Holmes et al., 1990*; *Kabsch et al., 1990*; *von der Ecken et al., 2016*; *Chou and Pollard, 2019*; *Zsolnay et al., 2020*; *Chou and Pollard, 2020*; *Belyy et al., 2020*; *Merino et al., 2018*; *Mentes et al., 2018*; *Mei et al., 2020*; *Ducka et al., 2010*; *Lee et al., 2007*; *Otomo et al., 2005*; *Oda et al., 2009*; *Egelman et al., 1982*; *Ali et al., 2022*; *Oda et al., 2020*; *Gong et al., 2022*). Although isoform-specific mechanisms with myosin motors and ABPs that drive functional distinction are widely described, they are poorly understood at the structural level.

To address how the structure contributes to the functional distinction of actin isoforms, we employed a combination of recombinant post-translationally modified actins and actins purified from native source to obtain pure, single-isotype preparations of individual actin isoforms to perform cryo-electron microscopy (cryo-EM) analyses. Specifically, we used our previously established Pick-ya actin method to recombinantly produce human ß-actin and γ-actin in an engineered *Pichia pastoris* strain that expresses the human N-acetyl transferase NAA80 and histidine methyl transferase SETD3 to ensure uniform Nt-acetylation and methylation of H72/H73, a conserved PTM profile of vertebrate actins (*Hatano et al., 2020*). Skeletal muscle α-actin and cardiac muscle α-actin were purified from

rabbit skeletal muscle and the left ventricle of a bovine heart, respectively. At the protein level, all actin isoforms are conserved across vertebrates, allowing us to compare our structures to previous structures of filamentous actin from other vertebrate species in the correct physiological context (*Figure 1—figure supplement 1*; *Pollard, 2016*; *Perrin and Ervasti, 2010*; *Dominguez and Holmes, 2011*). Our 2.99–3.38 Å resolution structures of filamentous actin isoforms show that the N-termini of bare muscle and nonmuscle actins have different orientations that contribute to distinct binding interfaces for myosin motors and likely other ABPs.

## Results

### High-resolution structures and general principles of actin isoforms

To determine the structural characteristics and differences between actin isoforms, we solved the high-resolution structures of single-isotype skeletal muscle α-actin (3.37 Å), cardiac muscle α-actin (3.07 Å), ß-actin (2.99 Å), and γ-actin (3.38 Å) in the filamentous state using cryo-EM (*Figure 1A–D*, *Figure 1—figure supplement 1*, *Table 1*). The structures show local resolutions ranging from 2.1 to 3.5 Å for skeletal muscle α-actin, 1.9–3.3 Å for cardiac muscle α-actin, 1.5–3.0 Å for ß-actin, and 2.1–4.4 Å for γ-actin (*Figure 1—figure supplement 2*). For all actin isoforms, the highest local resolutions were obtained in the central core region. Lower local resolutions were obtained in the most surface-exposed and flexible regions such as the N-terminus and the D-loop as expected. All actin structures were solved in the $Mg^{2+}$·ADP state (*Figure 1E*, *Figure 2*) and have been obtained without the use of stabilizing drugs that may interfere with the filament structure (*Diensthuber et al., 2011*; *Isambert et al., 1995*; *Zimmermann et al., 2015*).

Our cryo-EM maps allowed us to build unambiguous models of actin isoforms in which secondary structure information including the side chains, the nucleotide and associated cation ($Mg^{2+}$·ADP), and PTMs were apparent from the densities (*Figure 1E*, *Figure 1—figure supplement 2*). This allowed us to resolve the N-terminus of actin isoforms that is often disordered or missing in prior structures (*Kudryashov and Reisler, 2013*). For ß-actin and γ-actin, we could resolve the entire N-terminus starting from amino acids D1 and E1, respectively. For skeletal muscle α-actin and cardiac muscle α-actin, we could resolve the N-terminus starting from amino acids E4 and D2, respectively. The lack of resolvable density for the very first amino acid of cardiac muscle α-actin and the first three amino acids of skeletal muscle α-actin may be attributed to a nonuniform PTM pattern of native actin isoforms prepared from muscle compared to our recombinant nonmuscle actin isoforms with uniform PTM pattern in that the entire N-terminus region could be resolved (*Figure 1E*, *Figure 1—videos 1–5*).

Consistent with the high sequence conservation across actin isoforms (*Figure 1—figure supplement 1B–C*), our reconstructions show the characteristic double-stranded actin helix with a helical rise of ~27.6° to 28 Å and a helical twist of ~–166.5° to –168° (*Figure 1A–D*, *Figure 1—figure supplement 1A*) that has been observed in numerous previous structural studies, including previous high-resolution cryo-EM studies (*Holmes et al., 1990*; *Chou and Pollard, 2019*; *Mei et al., 2020*; *Oda et al., 2009*; *Egelman et al., 1982*; *Ali et al., 2022*; *Fujii et al., 2010*; *Galkin et al., 2015*). The actin filament itself is composed of G-actin (42 kDa) protomers that are oriented in the same direction (*Holmes et al., 1990*). Each protomer folds into four subdomains that are referred to as SD1–SD4 (*Figure 1*, *Figure 1—figure supplement 1A*; *Kabsch et al., 1990*). SD1 and SD2 form the outer domain, and SD3 and SD4 form the inner domain (*Figure 1*). This domain arrangement results in the formation of two clefts – the nucleotide-binding cleft and the barbed end groove (*Figure 1*, *Figure 1—figure supplement 1A*; *Pollard, 2016*; *Dominguez and Holmes, 2011*; *Merino et al., 2020*). The nucleotide-binding cleft between SD2 and SD4 harbors the active site that is occupied by $Mg^{2+}$·ADP in our structures (*Figure 1E*, *Figure 2*, *Figure 1—figure supplement 1A*). The barbed end groove between SD1 and SD3 represents a major binding interface for myosins and ABPs (*von der Ecken et al., 2016*). It further mediates longitudinal interfaces within the actin filament. SD2 and SD4 of an actin protomer are at the pointed end, and SD1 and SD3 are at the barbed end (*Figure 1—figure supplement 1A*; *Pollard, 2016*; *Dominguez and Holmes, 2011*; *Merino et al., 2020*). The longitudinal interface between two adjacent actin protomers involves the extended D-loop located in SD2 of one actin protomer that interacts with amino acids located in SD1 and SD3 of another

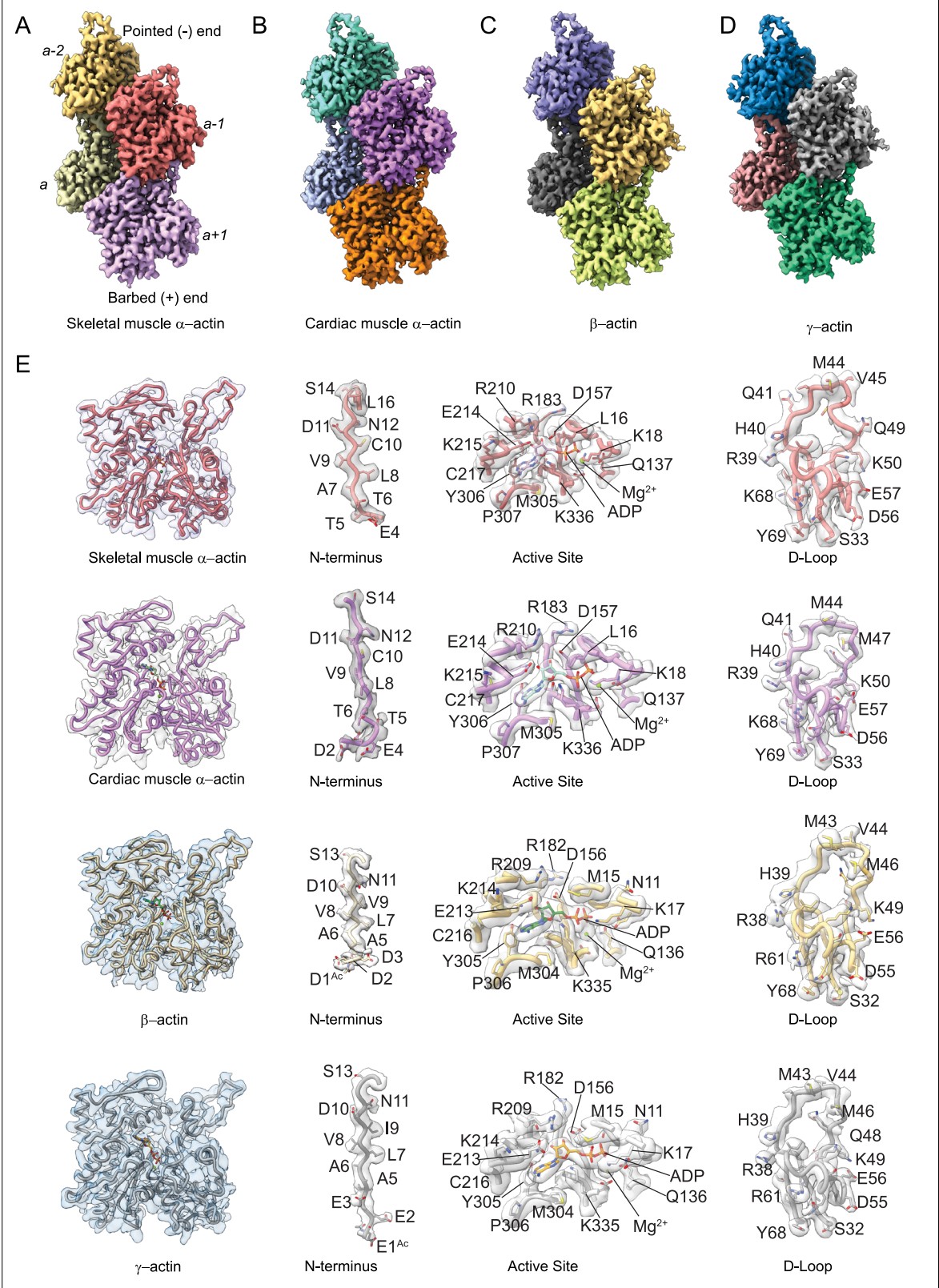

**Figure 1.** Cryo-electron microscopy (cryo-EM) filament structures of actin isoforms. (**A**) Helical reconstruction of skeletal muscle α-actin, (**B**) cardiac muscle α-actin, (**C**) β-actin, and (**D**) γ-actin. Views in (**B**–**D**) are according to (**A**). Four individual actin protomers in the filament are shown and denoted with italic numbers. The pointed (−) and barbed (+) ends are indicated. (**E**) Representative key regions of actin isoforms with corresponding cryo-EM densities in transparent surface representation are shown. The protein backbone and amino acid side chains are shown in licorice and stick

*Figure 1 continued on next page*

*Figure 1 continued*

representation, respectively. Throughout this work, amino acids are numbered according to the sequence of mature actin isoforms (*Figure 1—figure supplement 1C*).

The online version of this article includes the following video and figure supplement(s) for figure 1:

**Figure supplement 1.** Sequence conservation in actin isoforms.

**Figure supplement 2.** Image processing summary for actin isoforms.

**Figure supplement 3.** Methylation of H72/H73 in actin isoforms.

**Figure 1—video 1.** Structure of a skeletal muscle α-actin protomer with the corresponding cryo-electron microscopy (cryo-EM) density.
https://elifesciences.org/articles/82015/figures#fig1video1

**Figure 1—video 2.** Structure of a cardiac muscle α-actin protomer with the corresponding cryo-electron microscopy (cryo-EM) density.
https://elifesciences.org/articles/82015/figures#fig1video2

**Figure 1—video 3.** Structure of a β-actin protomer with the corresponding cryo-electron microscopy (cryo-EM) density.
https://elifesciences.org/articles/82015/figures#fig1video3

**Figure 1—video 4.** Structure of a γ-actin protomer with the corresponding cryo-electron microscopy (cryo-EM) density.
https://elifesciences.org/articles/82015/figures#fig1video4

**Figure 1—video 5.** Structures of the N-termini of actin isoforms with corresponding cryo-electron microscopy (cryo-EM) densities.
https://elifesciences.org/articles/82015/figures#fig1video5

protomer. Both the N- and C-terminus of the actin protomer are in SD1 (*Pollard, 2016*; *Dominguez and Holmes, 2011*; *Merino et al., 2020*).

## Similarities and differences between actin isoforms

The superposition of all actin isoform structures shows a root-mean-square deviation (RMSD) between 0.83Å to 1.04Å for Cα atoms, indicating an overall similar topology. No significant differences in the pitch of the actin helix were observed between our structures of actin isoforms, emphasizing their overall conserved filamentous structure in the absence of myosin motors or ABPs. Actin isoforms differ by conservative and nonconservative substitutions (*Figure 1—figure supplement 1C*) that contribute to their distinct biochemical and in vivo functions (*Blanchoin et al., 2014*; *Tondeleir et al., 2009*; *Vedula et al., 2021*). Overall, the amino acid sequence is more conserved among muscle actins than between muscle and nonmuscle actins (*Figure 1—figure supplement 1C*). A structural comparison of our cryo-EM reconstructions shows amino acid substitutions across isoforms with the positions of substituted amino acids highlighted (*Figure 3*). Actin isoforms show the largest divergence at the acidic N-terminus within SD1 (*Figure 1E*, *Figure 1—figure supplement 1C*, *Figure 3A*). Of note, amino acids 1–3 in our structure of skeletal muscle α-actin and the first amino acid in our structure of cardiac muscle α-actin are not resolved and therefore not shown in *Figure 3*. Other substitutions are within SD1 (*Figure 3B*, *Figure 3—figure supplement 1*), SD3 (*Figure 3C*, *Figure 3—figure supplement 1*), and SD4 (*Figure 3D*, *Figure 3—figure supplement 1*). There are no substitutions in SD2, the smallest and most flexible subdomain (*Kudryashov and Reisler, 2013*).

Amino acid substitutions at subdomain interfaces, such as the nucleotide-binding cleft active site of actin isoforms, are likely to influence protein function. To evaluate their possible impact on nucleotide coordination and the structural organization of the active site, we performed a comparative structural analysis. Our cryo-EM reconstructions show that the nucleotide-binding cleft active site is conserved between actin isoforms (*Figure 1E* and *Figure 2*). The densities for $Mg^{2+}$ and ADP were assigned without ambiguity and revealed interactions with amino acids located in SD2 (Q58/Q59 and Y68/Y69) and SD4 (E206/E207, R209/R210, K212/K213, and E213/E214) but also with amino acids located in SD1 (M15/L16, K17/K18, Q136/Q137, and Y336/Y337) and SD3 (D156/D157, M304/M305, Y305/Y306, and K335/K336; *Figure 2*).

While most of the interactions are polar and electrostatic, amino acid Y305/Y306 forms π-π interactions with the adenine ring of ADP. The superimposition of the nucleotide cleft active sites of our four structures of actin isoforms shows small differences in the positions of the nucleotide (RMSD ~0.44–0.47 Å) and the bound $Mg^{2+}$ (RMSD ~0.5–1.3 Å), especially in the position of the β-phosphate group relative to the α-phosphate group (*Figure 2*, *Figure 2—figure supplement 1*). The position of the $Mg^{2+}$ moves relative to the position of the β-phosphate group (*Figure 2—figure supplement 1*). The

**Table 1.** Data collection, image processing, and structure characteristics summary.

| Map | Skeletal muscle α-actin | Cardiac muscle α-actin | β-actin | γ-actin |
|---|---|---|---|---|
| **Data collection** | | | | |
| Microscope | FEI Titan Krios G3i | FEI Titan Krios G3i | FEI Titan Krios G3i | FEI Titan Krios G3i |
| Voltage (kV) | 300 | 300 | 300 | 300 |
| Detector | Gatan K3 | Gatan K3 | Gatan K3 | Gatan K3 |
| Automation software | EPU | EPU | EPU | EPU |
| Energy filter slit width (eV) | 20 | 20 | 20 | 20 |
| Recording mode | Super-resolution | Super-resolution | Super-resolution | Super-resolution |
| Magnification (nominal) | 81,000 | 81,000 | 81,000 | 81,000 |
| Movie micrograph pixel size (Å) | 0.891 | 0.891 | 0.891 | 0.891 |
| Total Dose rate ($e^-$/Å$^2$) | 65 | 60 | 50 | 65 |
| Defocus range (μm) | –0.5 to –2.5 | –0.5 to –2.5 | –0.5 to –2.5 | –0.5 to –2.5 |
| Spherical aberration (mm) | 0.01 | 0.01 | 0.01 | 0.01 |
| Movies | 2046 | 1444 | 1352 | 2952 |
| Total extracted particles | 261,195 | 657,300 | 279,120 | 1,249,379 |
| Total # of refined particles | 185,406 | 657,041 | 263,911 | 1,009,372 |
| **Reconstruction** | | | | |
| EMDB code | EMD-27548 | EMD-27549 | EMD-27572 | EMD-27565 |
| Box size | 350 | 256 | 256 | 256 |
| Symmetry | helical | helical | C1 | C1 |
| Map sharpening B-factor (Å$^2$) | –90 | –149 | –81 | –201 |
| Resolution (global) (Å) | 3.37 | 3.07 | 2.99 | 3.38 |
| **Structure building and validation** | | | | |
| PDB ID | 8 DMX | 8DMY | 8DNH | 8DNF |
| Model building | Coot | Coot | Coot | Coot |
| Refinement program | Phenix | Phenix | Phenix | Phenix |
| Refinement target | Real-space | Real-space | Real-space | Real-space |
| **RMSD from ideal values** | | | | |
| Bond length (Å) | 0.02 | 0.02 | 0.04 | 0.03 |
| Bond Angles (°) | 0.493 | 0.494 | 0.761 | 0.711 |
| Ramachandran favored (%) | 97.61 | 96.68 | 96.21 | 96.20 |
| Ramachandran allowed (%) | 2.39 | 3.32 | 3.79 | 3.46 |

*Table 1 continued on next page*

Table 1 continued

| Map | Skeletal muscle α-actin | Cardiac muscle α-actin | β-actin | γ-actin |
|---|---|---|---|---|
| Ramachandran outliers (%) | 0 | 0 | 0 | 0.34 |
| MolProbity Score | 1.42 | 1.42 | 1.72 | 1.72 |
| | | | | |
| **Structures Characteristics** | | | | |
| Species | Rabbit | Bovine | Human | Human |
| Amino acid resolved | 4–375 | 2–375 | 1–374 | 1–374 |
| PTMs resolved | H73 | H73 | D1/H72 | E1/H72 |

superimposition of the nucleotide-binding cleft active sites further shows small local rearrangements of side chains of conserved amino acids (*Figure 2—figure supplement 1*), including R182/R183 and K335/K336. Amino acids M15/L16 and M304/M305 form hydrophobic interactions in the nucleotide-binding cleft active site (*Figure 2*). Amino acid substitution M15/L16 between nonmuscle and muscle actin does not alter the overall topology of the nucleotide-binding site. Instead, the longer side chain of M15 in nonmuscle actins, located in a loop that protrudes into the nucleotide-binding cleft, acts as an extended lid that flanks the active site and shields the phosphate groups of ADP (*Figure 2*, *Figure 2—figure supplement 1*).

Near the nucleotide-binding cleft active site, amino acids C10 and V17 in muscle actins are substituted with V9/I9 and C16 in β/γ-actin (*Figure 3*, *Figure 1—figure supplement 1C*). These reciprocal amino acid substitutions maintain the overall oxidation-reduction environment within filamentous actin isoforms which is important for its dynamic properties and the interaction with some regulatory proteins (*Farah et al., 2011*; *Lassing et al., 2007*; *Wilson et al., 2016*; *Terman and Kashina, 2013*).

The analysis of interprotomer interfaces in our four structures of actin isoforms (*Figure 4*, *Figure 5*, *Figure 4—figure supplement 1*) showed that longitudinal interactions are mainly mediated by hydrophilic amino acids that are likely to enable interactions with water molecules that were recently shown to mediate interprotomer contacts within the filament core (*Figure 4*, *Figure 4—figure supplement 1*; *Reynolds et al., 2022*). Amino acid substitutions at the longitudinal interprotomer interface (also called long pitch helix interface) include L175/M176, T200/V201, Q224/N225, C271/A272, F278/Y279, and V286/I287. These substitutions may influence the stability of the promoters based on their ability to interact with the solvent and other protomer residues (*Reynolds et al., 2022*). The transverse interprotomer interface (also called short pitch helix interface) is formed through hydrophilic and hydrophobic interactions (*Figure 5*, *Figure 4—figure supplement 1*). In contrast to interactions at the longitudinal interface, most of the transverse interprotomer interactions are direct and not mediated by solvent. A single amino acid substitution (V286/I287) is present at the transverse interprotomer interface. This amino acid substitution is located near the intersection of the longitudinal and transverse interprotomer interfaces (*Figure 3—figure supplement 1*). Amino acid I287 is in hydrophobic contact with I208 and L242 in muscle actins and buries a surface area of ~113 Å², whereas the interaction between V286, I207, and L241 in nonmuscle actins is less hydrophobic and buries a surface area of ~83 Å² in the transverse interprotomer interface (*Figure 4—figure supplement 1C and D*). Interactions with the D-loop located in SD2 are conserved in our structures of actin isoforms, underlining the critical and conserved role of the D-loop to mediate interprotomer interactions. At the intersection of the longitudinal and transverse interprotomer interfaces, H39/40 and H172/H173 together with M43/M44 act as central anchors.

PTMs are not only essential for actin structure, function, and dynamics but also for their interaction with myosin motors and APBs (*Varland et al., 2019*; *Cook et al., 1993*; *Drazic et al., 2018*). Here, we focused on two widely documented and highly conserved PTMs of mature vertebrate actins that are important for actin structure and function: the post-translational acetylation of the N-terminus by N-terminal acetyltransferase NAA80 and methylation of H72/H73 by SET domain protein

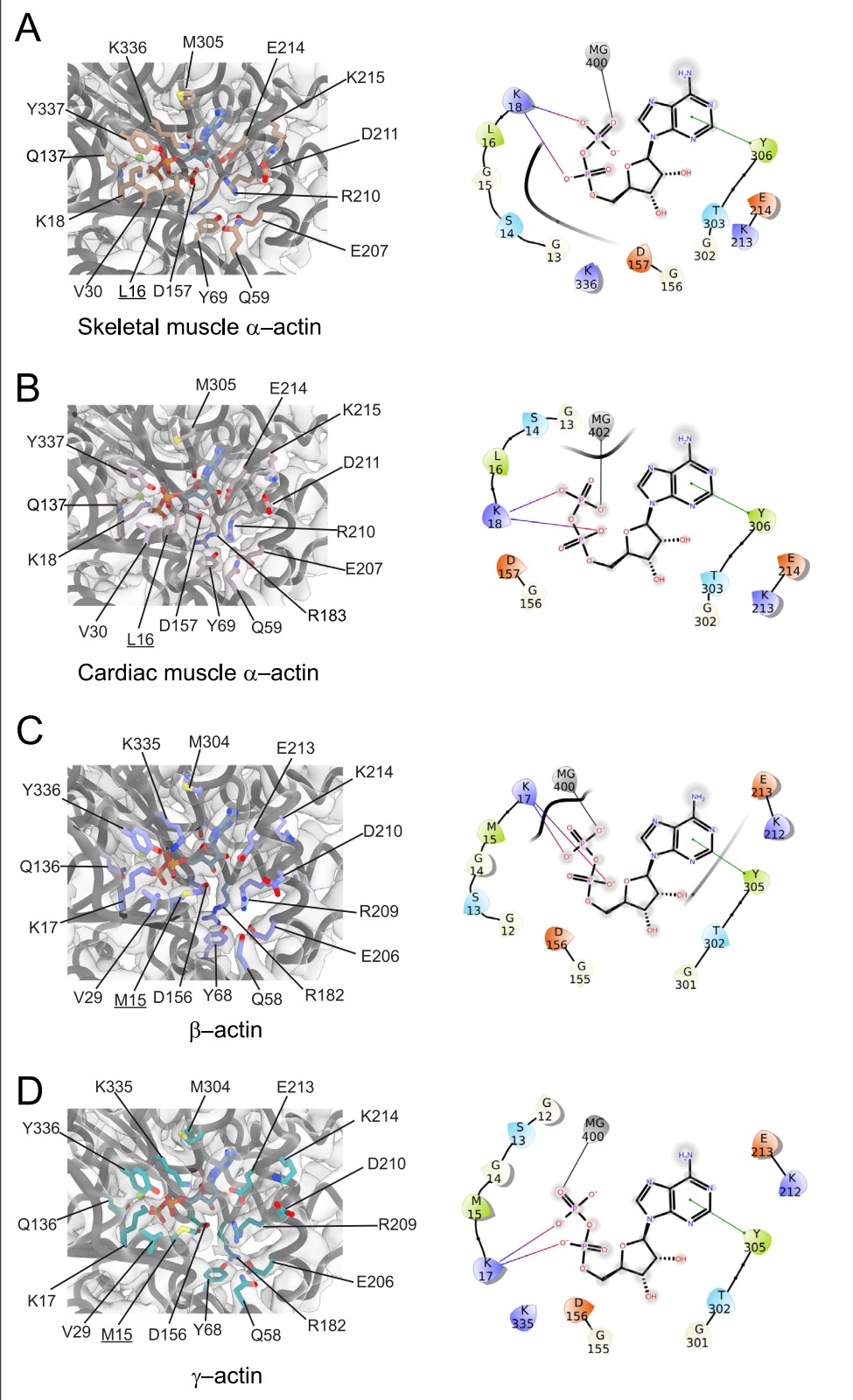

**Figure 2.** Conserved nucleotide-binding cleft active site in actin isoforms. (**A–D**) Coordination of Mg$^{2+}$·ADP in the nucleotide-binding cleft of skeletal muscle α-actin (**A**), cardiac muscle α-actin (**B**), β-actin (**C**), and γ-actin (**D**). Underlines indicate locations of amino acid substitutions between actin isoforms. The protein backbone and side chains are shown in licorice and stick representation, respectively. ADP is shown in cyan-colored stick

*Figure 2 continued on next page*

*Figure 2 continued*

representation. Electron densities for key amino acids in the nucleotide-binding cleft active site of actin isoforms are shown. Schematic representations of key interactions in the nucleotide-binding cleft active sites of the respective actin isoforms are shown in the right panel. The schematics are not drawn to scale.

The online version of this article includes the following figure supplement(s) for figure 2:

**Figure supplement 1.** Coordination of Mg²⁺·ADP in the nucleotide-binding cleft of actin isoforms.

3 (SETD3; *Kabsch et al., 1990*; *Terman and Kashina, 2013*; *Drazic et al., 2018*; *Wilkinson et al., 2019*; *Nyman et al., 2002*). Both PTMs are present in actin prepared from vertebrate tissues and our preparations of recombinant human β- and γ-actin produced in an engineered *Pichia pastoris* strain (*Hatano et al., 2020*). The quality of our density maps allowed us to resolve both key PTMs

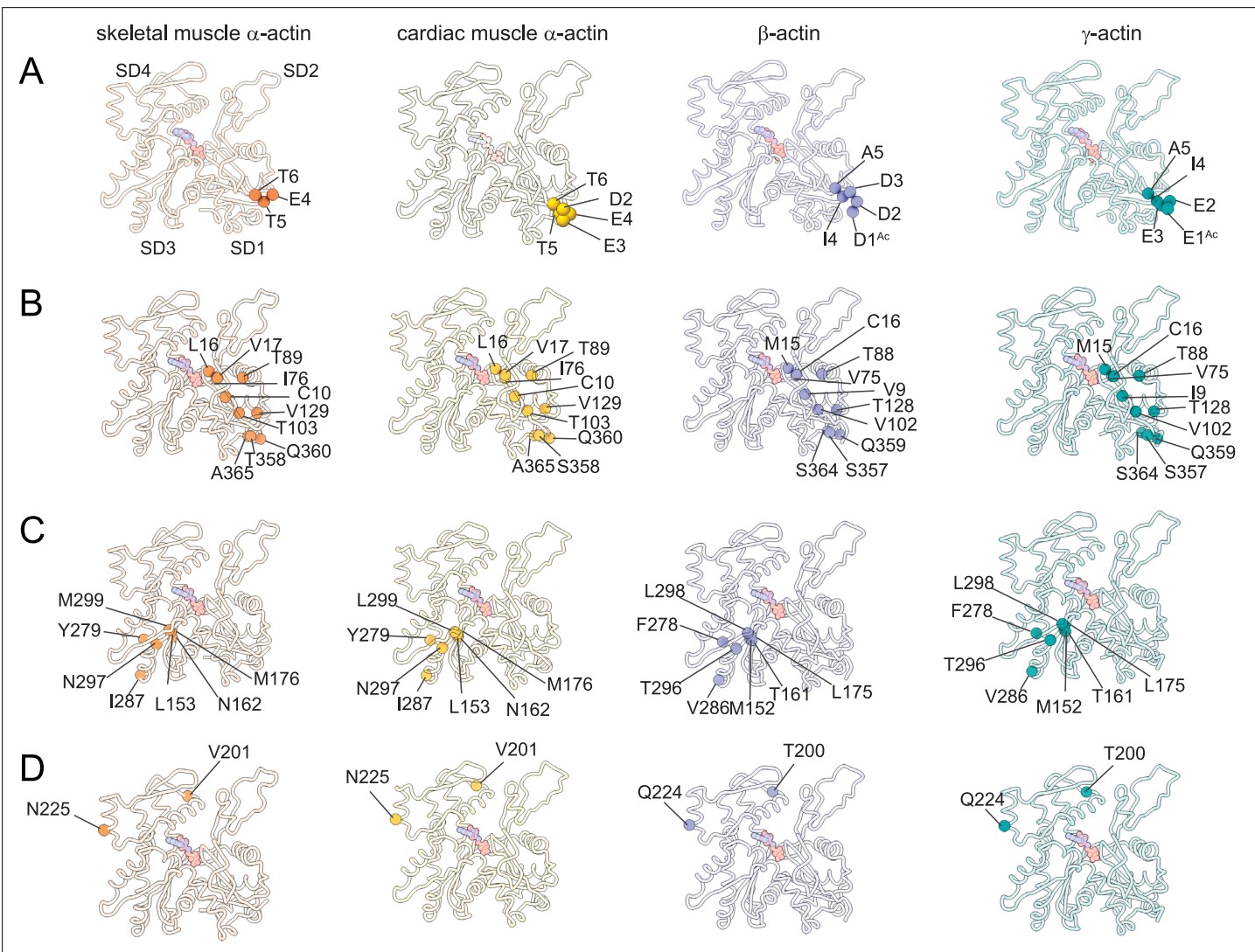

**Figure 3.** Similarities and differences between actin isoforms. (**A**) Sequence variations at the N-terminus located in SD1 of actin isoforms. (**B**) Sequence variations in SD1 of actin isoforms. (**C**) Sequence variations in SD3 of actin isoforms. (**D**) Sequence variations in SD4 of actin isoforms. SD2 is conserved between actin isoforms. The identical and nonidentical amino acids at sites of substitutions within the actin protomer across isoforms are shown for skeletal muscle α-actin (orange), cardiac muscle α-actin (yellow), β-actin (purple), and γ-actin (teal) as spheres. Note that the first three amino acids of skeletal muscle α-actin and the first amino acid of cardiac muscle α-actin are unresolved in our structures. The protein backbone is shown in licorice representation, and the substituted amino acids are shown in spheres representation.

The online version of this article includes the following figure supplement(s) for figure 3:

**Figure supplement 1.** Amino acid variations along the longitudinal and transverse axis of actin isoforms.

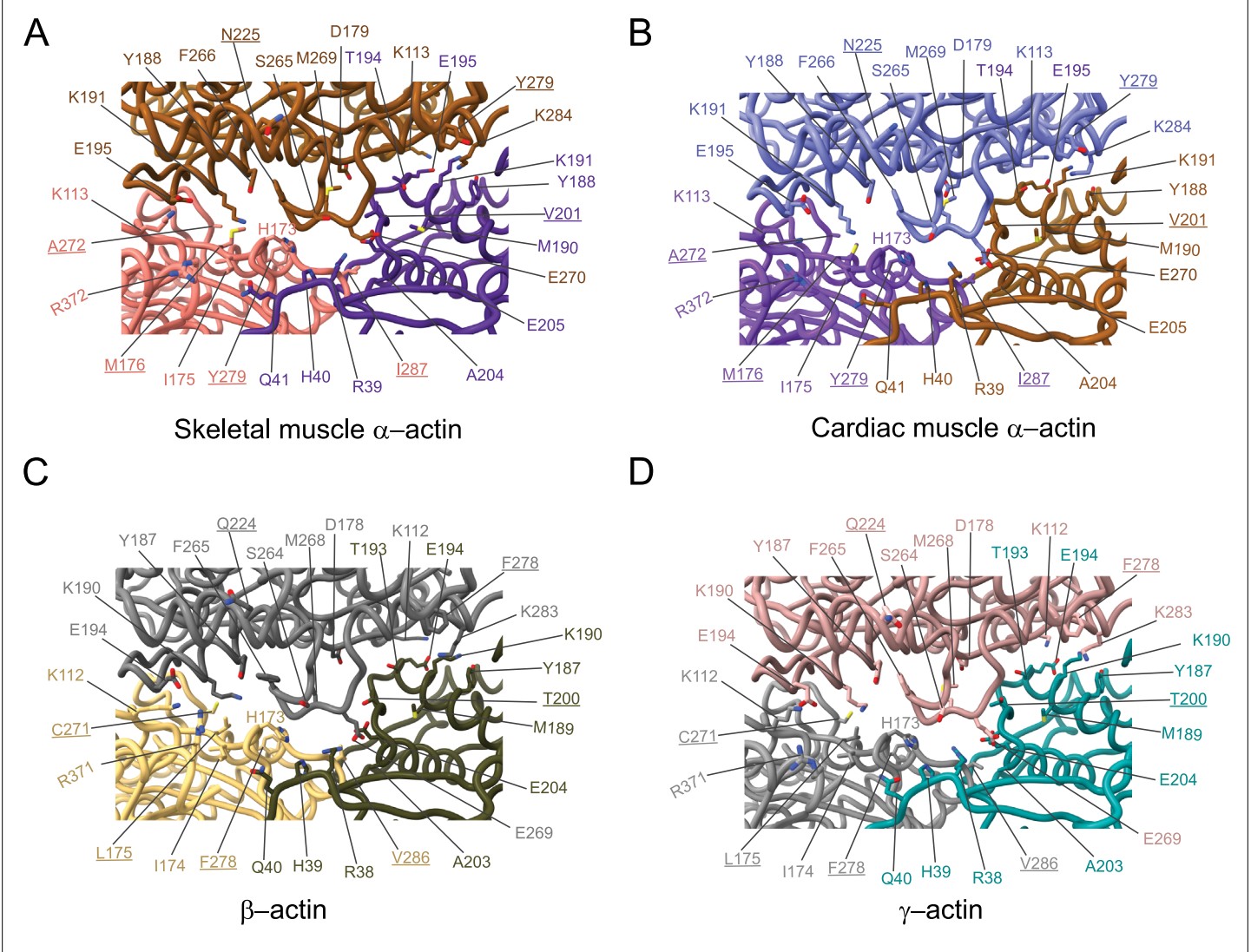

**Figure 4.** Comparative structural analysis of the longitudinal interprotomer interface. (**A–D**) Key residues at the interprotomer interface of skeletal muscle α-actin (**A**), cardiac muscle α-actin (**B**), β-actin (**C**), and γ-actin (**D**). Individual protomers in actin isoforms are oriented according to *Figure 4—figure supplement 1A*. Underlines indicate locations of amino acid substitutions between actin isoforms. The protein backbone and side chains are shown in licorice and stick representation, respectively.

The online version of this article includes the following figure supplement(s) for figure 4:

**Figure supplement 1.** Transverse and longitudinal interprotomer interfaces in actin isoforms.

(*Figure 1E*, *Figure 1—video 5*, *Figure 1—figure supplement 3*). Specifically, the presence of resolvable density allowed us to model the entire N-terminus including the Nt-acetylated D1 (D1^Ac) and the Nt-acetylated E1 (E1^Ac) in the cryo-EM reconstructions of β- and γ-actin (*Figure 1E*).

The acetylation site on the N-terminus is exposed on the filament surface and adds an additional negative charge to the already negatively charged N-terminus (*Figure 1—figure supplement 1C*). Like in previous cryo-EM structures of muscle α-actins, there are no resolvable densities for the very N-terminus, including Nt-acetylation, in our density maps of skeletal muscle α-actin and cardiac muscle α-actin (*Figure 1E*), possibly due to nonuniform PTM patterns of native, tissue-purified muscle actins. In addition to Nt-acetylation in β- and γ-actin, we could resolve the methylated H72/H73 (H72^Me/H73^Me) in all cryo-EM reconstructions of actin isoforms (*Figure 1—figure supplement 3*). The presence of both key PTMs in our cryo-EM reconstructions emphasizes that our recombinant human actins represent *bona fide* post-translationally processed, mature nonmuscle actins (*Hatano et al., 2020*).

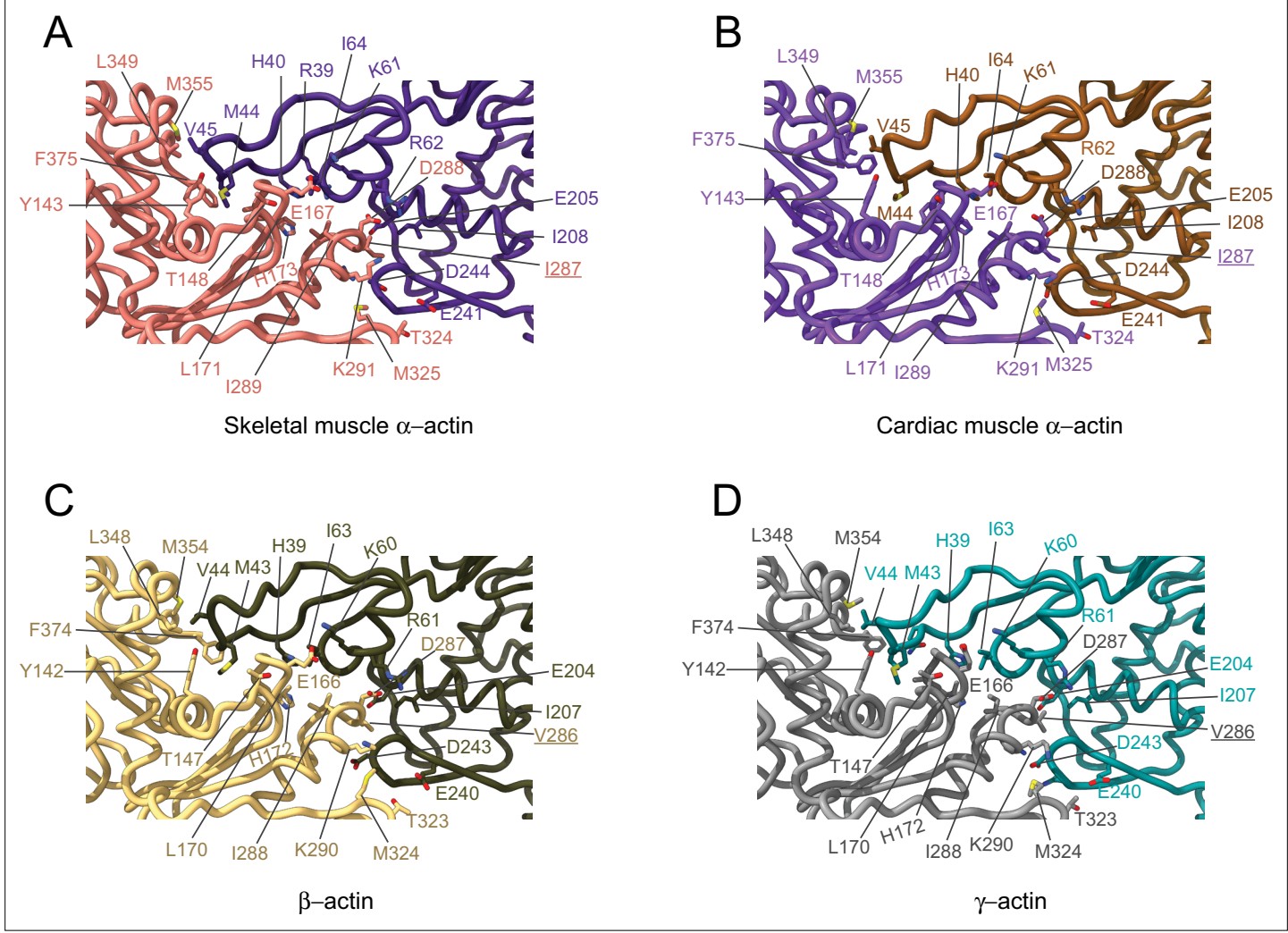

**Figure 5.** Comparative structural analysis of the transverse interprotomer interface. (**A–D**) Key residues at the interprotomer interface of skeletal muscle α-actin (**A**), cardiac muscle α-actin (**B**), β-actin (**C**), and γ-actin (**D**). Individual protomers in actin isoforms are oriented according to *Figure 4—figure supplement 1B*. Underlines indicate locations of amino acid substitutions between actin isoforms. The protein backbone and amino acid side chains are shown in licorice and stick representation, respectively.

## Myosin modulates actin filament structure

Myosin motors bind actin in a nucleotide-dependent, reversible manner to generate force and motion (*Heissler and Sellers, 2016a*; *Sellers, 2000*). The myosin enzymatic cycle can be categorized into states with weak (ATP and ADP·Pi) and strong (ADP and nucleotide-free [rigor]) actin affinity (*De La Cruz and Ostap, 2009*; *Heissler and Sellers, 2016b*). In the strong affinity states, a binding interface is established between actin and the myosin motor domain (*von der Ecken et al., 2016*; *Doran and Lehman, 2021*; *Lorenz and Holmes, 2010*). To determine whether myosin binding to actin may modulate actin filament structure, we compared our structures of filamentous bare actin isoforms with previous high-resolution cryo-EM structures of myosin-bound actins. We compared the structure of rigor nonmuscle myosin-2C (NM2C, PDB ID: 5JLH) bound to γ-actin, the structure of rigor myosin-1B bound to skeletal muscle α-actin (M1B, PDB ID: 6C1H), and the structure of Mg$^{2+}$·ADP.M1B bound to skeletal muscle α-actin (PDB ID: 6C1G) with our structures of bare actin isoforms (*Figure 6*). We also compared the structure of rigor cardiac myosin-2 bound to the cardiac thin filament (cardiac muscle α-actin decorated with tropomyosin and troponin, PDB ID: 7JH7; *Figure 6—figure supplement 1*) with our structure of bare cardiac muscle α-actin. The actomyosin structures were selected since they represent distinct high-affinity binding states (Mg$^{2+}$·ADP versus rigor) and classes of myosin motors

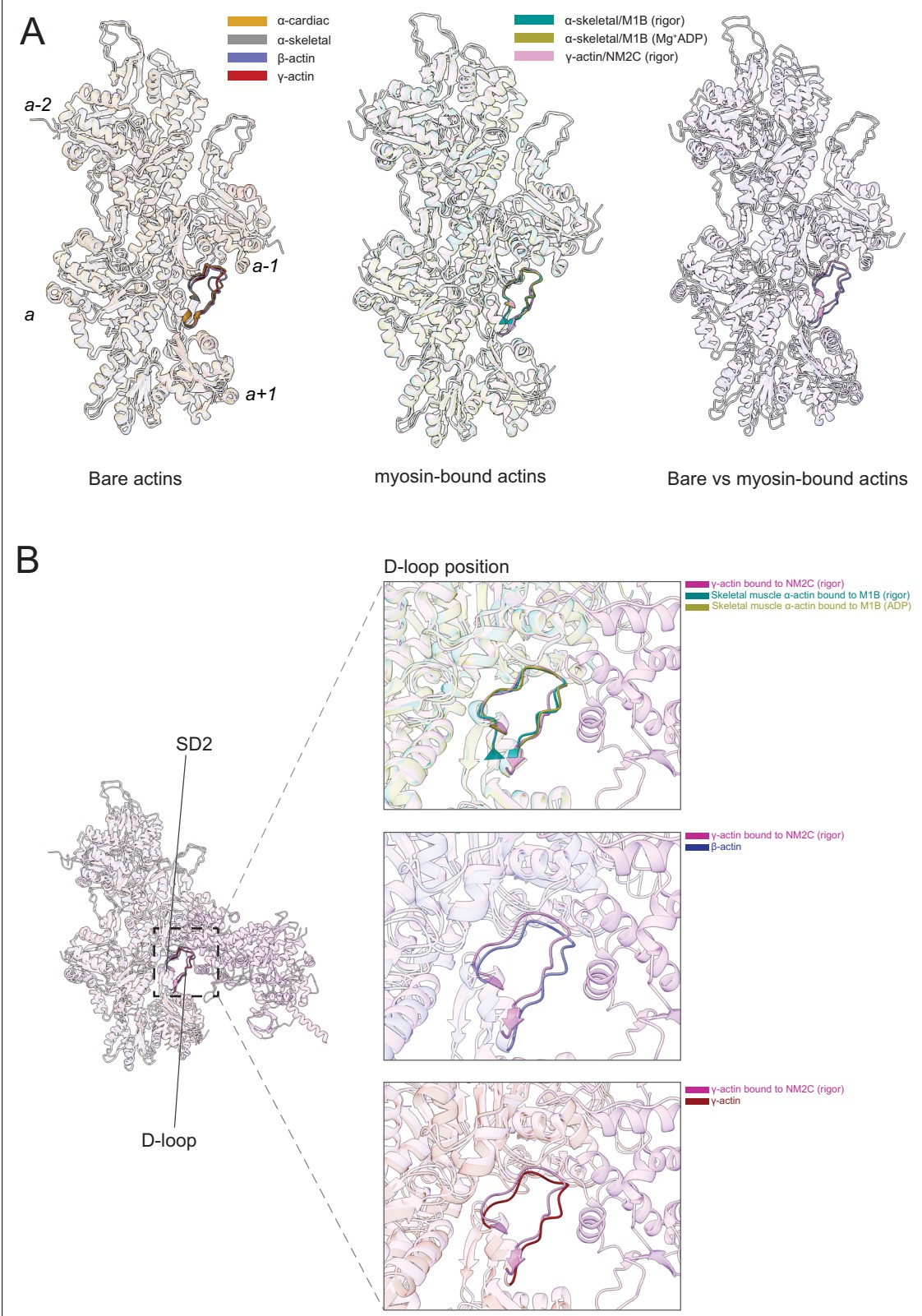

**Figure 6.** The actomyosin interface. (**A**) Superimposition of bare actin isoform structures in the Mg$^{2+}$·ADP state (left), superimposition of myosin-bound actin isoforms structures (middle), and overlap of bare versus myosin-bound actin structures (right) are shown. (**B**) Zoomed-in view of the actomyosin interface at the D-loop region. For clarity, only D-loops involved in the binding of myosins are highlighted in the respective dark colors. The offset between the structures in the lower two panels is caused by a conformational change of SD2 in myosin-bound compared to bare actin structures.

*Figure 6 continued on next page*

Figure 6 continued

The online version of this article includes the following video and figure supplement(s) for figure 6:

**Figure supplement 1.** The actomyosin interface of the cardiac thin filament compared to bare cardiac muscle α-actin.

**Figure 6—video 1.** Conformational changes of the D-loop in bare and myosin-bound actin.

https://elifesciences.org/articles/82015/figures#fig6video1

with different enzymatic output (*Mentes et al., 2018*; *Heissler and Sellers, 2016b*; *Heissler and Manstein, 2011*).

Myosin binding to actin filaments causes subtle conformational changes in the filament (*Figure 6*). Superimposition of the actin structures showed that the Cα of bare and myosin-bound actin protomers deviate with an RMSD of ~1.25–3 Å. These changes are similar for myosins from different classes and independent of the nucleotide state of the respective motor domain (*Figure 6A*). The superposition of structures of bare actin isoforms and known structures of myosin-bound actins shows that SD2 adopts a different conformation (*Figure 6A and B*, *Figure 6—figure supplement 1*, *Figure 6—video 1*). The conformation of the D-loop, located in SD2, differs with a Cα RMSD of ~0.95–1.25 Å (*Figure 6A and B*). Superimposition of bare γ-actin with γ-actin/NM2C complex shows the subtle inward movement of the D-loop with a Cα RMSD of ~1.25 Å in the myosin-bound state, suggesting that myosin-binding induces subtle changes in the barbed end groove that do not change the separation between SD1/SD3 and SD2/SD4. The inward movement of the D-loop (RMSD ~0.91 Å) is also observed in the recent structure of cardiac myosin bound to the thin filament compared to our structure of bare cardiac muscle α-actin (*Figure 6—figure supplement 1*), suggesting that myosin and not the regulatory proteins tropomyosin and troponin drive this structural change. The analysis of the actomyosin interface also shows that the D-loop conformation does not change significantly with respect to the nucleotide states (rigor versus ADP) of the compared myosins (*Figure 6B*).

Next, we compared the position of the actin N-terminus region and its interaction with myosin motors. The superimposition of bare and myosin-bound actin structures shows that the N-termini of skeletal muscle α-actin points in a different direction compared to those from ß-actin and γ-actin (*Figure 7A*). The actin N-terminus is positioned closer to the filament surface in myosin-bound structures compared to our bare actin structures (*Figure 7*, *Figure 6—video 1*, *Figure 7—video 1*). The negatively charged N-termini of bare ß-actin and γ-actin are near loop-2, a major element of the actomyosin interface that is rich in positively charged amino acids and variable in length (*Heissler and Manstein, 2011*; *Joel et al., 2001*; *Murphy and Spudich, 1999*; *Uyeda et al., 1994*).

This interface is likely to represent one of the initial contact sites upon the formation of a transient intermediate between both proteins during the formation of the actomyosin complex. Nt-acetylation of actin likely enhances long-range electrostatic interactions between both proteins that subsequently trigger allosteric structural changes that result in the formation of the actomyosin interface as suggested previously (*Abe et al., 2000*). Furthermore, we observed a cluster of positively charged amino acids (R661, R663, and R664) in loop-2 of NM2C that may interact with either the acetylated N-terminus of ß-actin (D1$^{Ac}$, D2, and D3) or the acetylated N-terminus of γ-actin (E1$^{Ac}$, E2, and E3; *Figure 7B–C*) to stabilize the interface.

We also showed that a shorter loop-2, as it is found in M1B, is less efficient in stabilizing the actin N-terminus due to geometric constraints that limit its ability to pull the N-terminus closer to the filament surface in the strong binding states compared to a longer loop-2 as it is found in NM2C (*Figure 7A*). This suggests that distinct actin-myosin interfaces are formed between actin isoforms and myosin motor proteins that may determine the strength of the interaction and biochemical outputs.

## Discussion

Despite many elegant, high-resolution cryo-EM studies of filamentous actin in the absence and presence of myosin motors and ABPs, our understanding of the structural mechanisms that underlie the nuanced interactions of actin isoforms with interacting proteins that have been previously reported in cells and in vitro remain largely elusive (*Müller et al., 2013*; *Chen et al., 2017*; *Dugina et al., 2009*). This knowledge however is critical to understanding how protein binding modulates actin filament structure and networks that in turn may be recognized by and guide other interacting proteins to

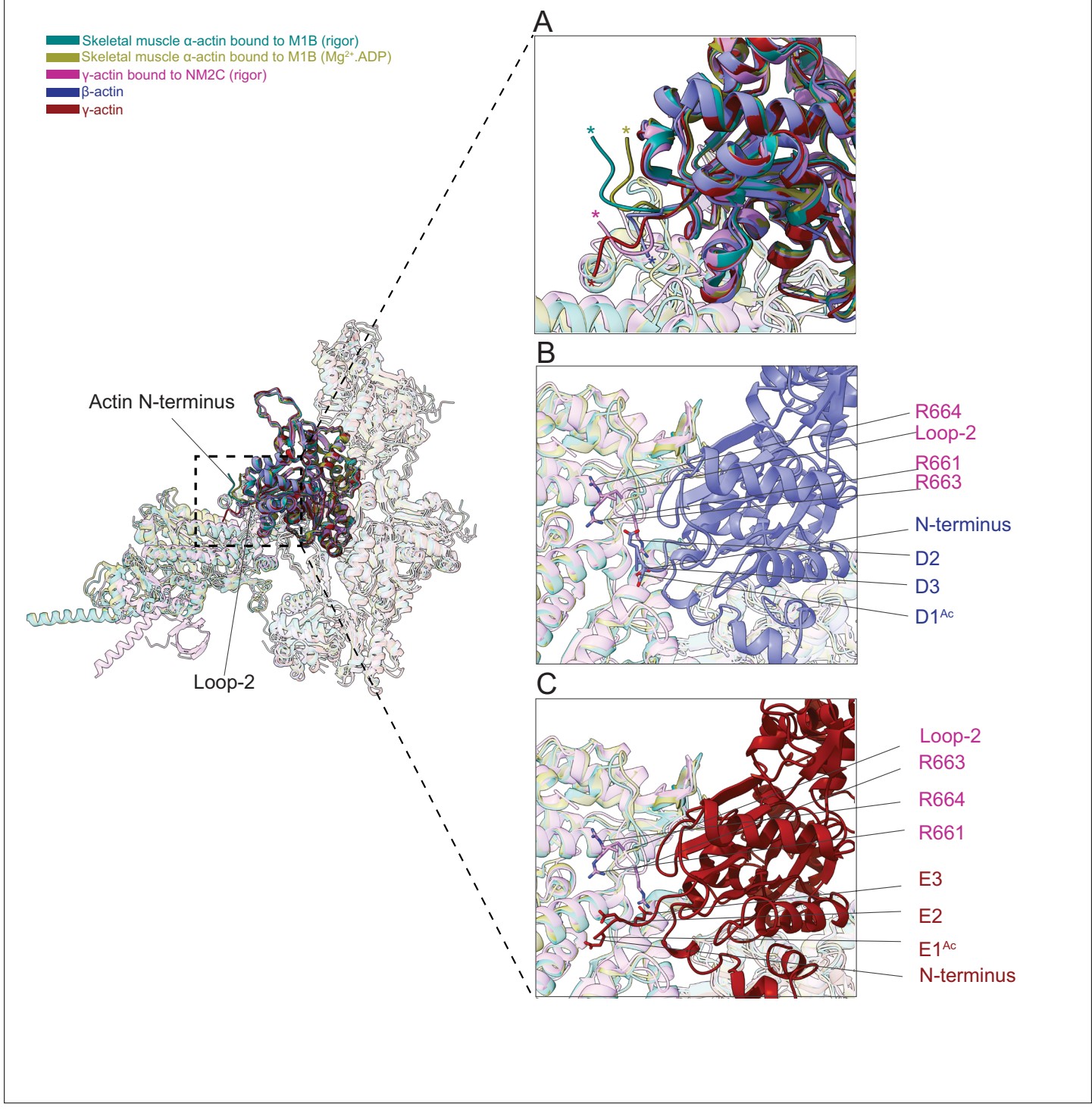

**Figure 7.** The N-terminus of actin interacts with loop-2 of myosins. (**A**) Close-up view of the N-termini of bare β-actin (blue), bare γ-actin (red), myosin-bound actin isoforms M1B (green, lime), and NM2C (pink). The N-termini are indicated with an asterisk. (**B**) A cluster of positively charged amino acids in the loop-2 of myosins is in close proximity to amino acids D1Ac to D3 in the N-terminus of bare β-actin (blue) and, (**C**) a cluster of positively charged amino acids in the loop-2 of myosins is in close proximity to amino acids E1Ac to E3 in the N-terminus of bare γ-actin (red).

The online version of this article includes the following video for figure 7:

**Figure 7—video 1.** Conformational changes at the actomyosin interface.

https://elifesciences.org/articles/82015/figures#fig7video1

create a variety of actin landscapes in cells (*Harris et al., 2020*; *Lappalainen, 2016*; *Breitsprecher et al., 2012*; *Damiano-Guercio et al., 2020*; *Jansen and Goode, 2019*; *Chen et al., 2020*).

To further our understanding of the sequence-structure relationship of actin isoforms, we solved four high-resolution cryo-EM structures, including the structure of bare and isotypic filamentous skeletal muscle α- actin and the new structures of bare cardiac muscle α-actin, ß-actin, and γ-actin. The structures of single isotype, mature actins allowed us to analyze the similarities and differences between actin isoforms and their unique modes of interactions with myosin motors.

Actin isoforms are subject to extensive PTM mechanisms that may drive structural changes, distinct biochemistries, and cellular activities (*Kashina, 2020*; *Vedula et al., 2021*; *Terman and Kashina, 2013*). Prevalent PTMs of actin isoforms include acetylation and methylation. Nt-acetylation and H72/H73 methylation are highly conserved in vertebrate actins and therefore expected to be present in most if not all previous structures of filamentous F-actin. However, the methylated H72/H73 has been only recently resolved in the structure of skeletal muscle α-actin (*Chou and Pollard, 2019*). The high resolution of our structures allowed us to resolve the methylated H72/H73. Furthermore, we could resolve the acetylated N-termini in filamentous ß-actin and γ-actin.

The overall structure of actin isoforms is conserved, as expected from the high-sequence conservation (*Figure 1—figure supplement 1*). However, each structure shows subtle but defining differences. Minor differences were observed near the nucleotide-binding cleft (*Figure 2*, *Figure 2—figure supplement 1*) that may contribute to the reported differences in the biochemistries of actin isoforms (*Chen et al., 2021*; *Bergeron et al., 2010*). The comparison of longitudinal and transverse inter-protomer interfaces revealed that the transverse interface is, with only one amino acid substitution (V286/I287), more conserved compared to the longitudinal interprotomer interface which features several amino acid substitutions (L175/M176, T200/V201, Q224/N225, C271/A272, F278/Y279, and V286/I287; *Figure 4*, *Figure 4—figure supplement 1C and D*, *Figure 5*). Furthermore, amino acid substitution V286/I287 is located at the intersection of the longitudinal and transverse interprotomer interfaces (*Figure 3—figure supplement 1*). In addition, our structures show the presence of conserved amino acids (H39/H40, H172/H173, and M43/M44) at the intersection of the longitudinal and transverse interprotomer interfaces. These residues were previously shown to be highly susceptible to oxidative stress caused by reactive oxygen species and suggest a possible implication for the filament stability of actin isoforms (*Varland et al., 2019*). The absence of amino acid substitutions in SD2 and conserved interactions between the SD2 D-loop and amino acids of SD1 and SD3 suggests selective pressure to maintain this critical structural element. We also noticed that the D-loop of actin isoforms is located closer to the filament surface in myosin-bound structures compared to structures of bare actin filaments.

In contrast, the recent high-resolution structures of rigor and Mg$^{2+}$·ADP.myosin-15 bound to skeletal muscle α-actin suggest flexibility in the D-loop in the rigor state and a rigid D-loop near the actin surface in the Mg$^{2+}$·ADP state. The flexible D-loop has been proposed to support actin nucleation activity, whereas the rigid D-loop has been proposed to limit the actin nucleation activity of this myosin (*Gong et al., 2022*).

The most prominent structural change between bare actin isoforms corresponds to the N-terminus location (*Figure 1*, *Figure 7*). Given that the sequence of actin isoforms differs the most at the N-terminus (*Figure 1—figure supplement 1C*) and the well-established central role of the N-terminus in the interactions with myosin motors and ABPs, we propose that these structural changes may dictate different biochemical interactions. For example, by comparing the structures of bare skeletal muscle α-actin with the γ-actin/NM2C complex, a previous study suggested a large-scale pulling mechanism in which the actin N-terminus is pulled toward the actomyosin interface in the rigor state (*von der Ecken et al., 2016*). Based on our structures that show different conformations of the N-terminus between muscle and nonmuscle actins, we suggest that instead of a large conformational change through a pulling mechanism, actin isoform-specific interfaces with myosin motors are formed that involve a more subtle movement of the N-terminus. These different interfaces together with the subtle differences in the amino acid sequence of the N-terminus region are likely to contribute to the extent actin isoforms can activate the enzyme function of myosin motors in vitro and in cells (*Müller et al., 2013*; *Lappalainen, 2016*). The presence of low levels of nonmuscle actins and nonmuscle myosins in muscle suggests that motors, and likely other interacting partners, can interact with multiple actin isoforms in vivo. In fact, most cells maintain multiple actin isoforms and a defined complement of

myosin motors (*Kashina, 2020*; *Otey et al., 1987*; *Sellers, 2000*; *Heissler and Sellers, 2016b*). This cell type-specific mixing and matching is likely to fine-tune the enzymatic output of myosin motors for specialized functions (*Kee et al., 2009*; *Müller et al., 2013*; *Sun et al., 2020*). For example, previous in vitro experiments with recombinant proteins revealed isoform-specific differences in the interaction between muscle and nonmuscle actin isoforms and individual myosin motors (*Müller et al., 2013*). Notably, predominantly nonsarcomeric myosins have a faster ATPase activity and in vitro sliding velocity when assayed with nonmuscle actins compared to muscle actin, suggesting selective fine-tuning of the functional competencies of myosin-actin combinations with implications for cell function (*Müller et al., 2013*). These in vitro observations are supported by our structural data that suggest that the different orientations of the N-termini of actin isoforms contribute to the formation of different binding interfaces with myosin motors (*Figure 7*). Previous work also revealed differences in the extent to which the highly conserved β-actin and γ-actin isoforms (*Figure 1—figure supplement 1*) can activate the kinetic and functional activity of the same motor in vitro (*Müller et al., 2013*). Based on our structural and previous biochemical studies, we speculate that the biochemical properties of the very N-terminus of actin isoforms are key contributors to the formation of isoform-specific interfaces with myosin motors (*Müller et al., 2013*; *Cook et al., 1993*; *Abe et al., 2000*). For instance, the very N-termini of mature β-actin (DDD) and γ-actin (EEE) have different pKa values, resulting in a higher negative electron charge density in β-actin compared to γ-actin that is further increased by Nt-acetylation in both isoforms. These differences are likely to contribute to distinct interactions with the positively charged loop-2 and other actin-binding elements in the motor domain of myosin family members during the formation of the actomyosin interface. Actin isoform-specific interfaces may be further diversified by differential N-terminal processing of β-actin, PTMs, and alternatively spliced myosin motors (*Varland et al., 2019*; *Müller et al., 2013*; *Arnesen et al., 2018*; *Terman and Kashina, 2013*; *Heissler and Manstein, 2011*; *Sheff and Rubenstein, 1989*; *Solomon and Rubenstein, 1985*). For example, the Nt-acetylated N-termini of actin isoforms would likely be more efficient to establish electrostatic interactions with the positively charged loop-2 of myosin, while the addition of a positive charge to the N-terminus of β-actin through arginylation may weaken the interaction.

While we focused our comparative structural analysis on the interaction between actin isoforms with myosin motors, the actin N-terminus has been shown previously to interact with numerous ABPs (*Arnesen et al., 2018*; *Cook et al., 1993*; *Abe et al., 2000*). Therefore, we speculate that the proposed isoform-specific interaction mechanisms that stem from the different conformations of the N-termini extend to ABPs. Together, we present structural evidence that the binding of myosin motors modulates actin filament structure and propose a possible mechanism for the formation of actin isoform-specific interactions with binding proteins by the conformation of its N-terminus (*Figure 7*).

In conclusion, we present direct evidence for the structural divergence of actin isoforms that underlies their nuanced interactions with myosin motors. Our work serves as a strong foundation for our understanding of the sequence-function relationship of actin isoforms. By adding our cryo-EM structures of single isotype, mature actin isoforms to the collection of previous structures of the bare and decorated actin filaments, we provide a comprehensive understanding of the remarkable diversity of actin isoforms that is reflected in their biological activities in health and disease.

## Materials and methods

**Key resources table**

| Reagent type (species) or resource | Designation | Source or reference | Identifiers | Additional information |
|---|---|---|---|---|
| Strain, strain background (*P. pastoris*) | *P. pastoris* transformants | https://journals.biologists.com/jcs/article/131/8/jcs213827/57192/Rapid-production-of-pure-recombinant-actin | | *Hatano et al., 2018*; *P. pastoris* transformants used to prepare recombinant b-actin (*H. sapiens*) and g-actin (*H. sapiens*) |
| Biological sample (*O. cuniculus*) | Skeletal muscle | Pel-Freez | 41995 | Muscle acetone powder used for the preparation of native skeletal muscle a-actin |
| Biological sample (*B. taurus*) | heart | Local butcher | | Left ventricle used for the preparation of native cardiac muscle a-actin |

*Continued on next page*

*Continued*

| Reagent type (species) or resource | Designation | Source or reference | Identifiers | Additional information |
|---|---|---|---|---|
| Other (*O. cuniculus*) | skeletal muscle a-actin | This paper | UniProt ID: P68135 | Peptide, protein reagent, prepared from muscle acetone powder |
| Other (*B. taurus*) | cardiac muscle a-actin | This paper | UniProt ID: Q3ZC07 | Peptide, protein reagent, prepared from bovine heart |
| Peptide, recombinant protein (*H. sapiens*) | b-actin | This paper | UniProt ID: P60709 | Prepared from *P. pastoris* transformants |
| Peptide, recombinant protein (*H. sapiens*) | g-actin | This paper | UniProt ID: P63261 | Prepared from *P. pastoris* transformants |
| Other | Amicon 30 kDa MWCO centrifugal filters | Millipore Sigma | UFC903008 | Protein concentrators |
| Other | C-flat Au 1.2/1.3 grids | Electron Microscopy Sciences | CF313-50-Au | Electron microscopy grids |
| Software, algorithm | EPU | Thermo Fisher Scientific | | Software for cryo-EM data acquisition |
| Software, algorithm | cryoSPARC | https://doi.org/10.1038/nmeth.4169 | | *Punjani et al., 2017* |
| Software, algorithm | MotionCor2 | https://doi.org/10.1038/nmeth.4193 | | *Zheng et al., 2017* |
| Software, algorithm | Coot | https://doi.org/10.1107/S0907444910007493 | | *Emsley et al., 2010* |
| Software, algorithm | PHENIX | https://doi.org/10.1107/S0907444909052925 | | *Adams et al., 2010* |
| Software, algorithm | Chimera | https://doi.org/10.1002/jcc.20084 | | *Pettersen et al., 2004* |
| Software, algorithm | MolProbity | https://doi.org/10.1002/pro.3330 | | *Williams et al., 2018* |

## Protein production and purification

### Native actins

Rabbit skeletal muscle α-actin (UniProt ID: P68135) was prepared from acetone powder (Pel-Freez Biologicals, Rogers, AR, USA) as described earlier (*Heissler et al., 2015*). Cardiac muscle α-actin (UniProt ID: Q3ZC07) was prepared from the left ventricle of a bovine heart (local butcher) as described for skeletal muscle α-actin. At the amino acid level, both proteins are identical to the respective human proteins.

### Recombinant actins

*P. pastoris* transformants for human β-actin (UniProt ID: P60709) and human γ-actin (UniProt ID: P63261) were stored at –80°C and were revived on YPD (Yeast extract, Peptone, Dextrose) solid media plates at 30°C. Cells were inoculated into 200 mL Minimal Glycerol (MGY) liquid media composed of 1.34% yeast nitrogen base without amino acids (Millipore Sigma, St. Louis, MO, USA), 0.4 mg/L biotin, and 1% glycerol and cultured at 30°C, 220 rpm. The culture medium was diluted to 6 L with fresh MGY media, and cells were further cultured at 30°C, 220 rpm in six 2 L flasks until the optical density at 600 nm (OD600) reached around 1.5. Cells were pelleted down by centrifugation (10,628 g at 25°C for 5 min, F9–6x1,000 LEX rotor, Thermo Fisher Scientific, Waltham, MA, USA). The cells were washed once with sterilized water and re-suspended into 6 L MM composed of 1.34% yeast nitrogen base without amino acids (SIGMA Y0626), 0.4 mg/L biotin, and 0.5% methanol. Cells were cultured in twelve 2 L baffled flasks (500 mL for each) at 30°C, 220 rpm for 1.5–2 days. 0.5% methanol was fed every 24 hr during the culture. Cells were pelleted down by centrifugation (10,628 g at 25°C for 5 min, F9–6x1,000 LEX rotor, Thermo Fisher Scientific, Waltham, MA, USA). Cells were washed once with water and suspended in 75 mL of ice-cold water. The suspension was dripped into a liquid nitrogen bath and stored at –80°C. 50 g cell suspension was loaded into a grinder tube (SPEX SamplePrep, Metuchen, NJ, USA) pre-cooled with liquid nitrogen. Cells were grinded in a liquid nitrogen bath of a Freezer mill (SPEX SamplePrep, Metuchen, NJ, USA). The duration of the grinding was 1 min with 14 cycles per second. The grinding procedure was repeated 30 times at 1 min intervals. Liquid nitrogen was re-filled every 10 times of the grinding. The resulting powder was kept on dry ice until the next

step. The powder was kept at room temperature until it started to melt. Then, it was resolved in an equal amount of 2× binding buffer composed of 20 mM imidazole (pH 7.4), 20 mM HEPES (pH 7.4), 0.6 M NaCl, 4 mM MgCl$_2$, 2 mM ATP (pH 7.0), 2× concentration of protease inhibitor cocktail (Roche cOmplete, EDTA free, Millipore Sigma, St. Louis, MO, USA), 1 mM phenylmethylsulfonyl fluoride (PMSF), and 7 mM beta-mercaptoethanol. The lysate was sonicated on ice (3 min, 5 s pulse, 10 s pause with 60% amplitude, QSONICA SONICATORS, Newtown, CT, USA) until all aggregates were resolved. The lysate was centrifuged at 4°C (3220 g for 5 min, A-4–81 rotor, Eppendorf, Enfield, CT, USA) to remove intact cells and debris. The insoluble fraction was removed by high-speed centrifugation at 4°C (25,658 g for 30 min, A23–6x100 rotor; Thermo Fisher Scientific, Waltham, MA, USA). The supernatant was filtrated with a Filtropur BT50 0.2 µm bottle top filter (SARSTEDT, Nümbrecht, Germany) and incubated with 6 mL Nickel resin (Thermo Fisher Scientific, Waltham, MA, USA) at 4°C for 1 hr. The resin was pelleted down by centrifugation at 4°C (1258 g for 5 min, A-4–81 rotor, Eppendorf, Enfield, CT, USA) and washed with ice-cold 50 mL binding buffer composed of 10 mM imidazole (pH 7.4), 10 mM HEPES (pH 7.4), 300 mM NaCl, 2 mM MgCl$_2$, 1 mM ATP (pH 7.0), and 7 mM beta-mercaptoethanol for four times. The resin was washed five times with a G-buffer composed of 5 mM HEPES (pH 7.4), 0.2 mM CaCl$_2$, 0.01 w/v% NaN$_3$, 0.2 mM ATP (pH 7.0), and 0.5 mM dithiothreitol. The resin was suspended in an ice-cold 40 mL G-buffer with 5 µg/mL TLCK-treated chymotrypsin (Millipore Sigma, St. Louis, MO, USA) and incubated overnight at 4°C. The chymotrypsin was inactivated by 1 mM PMSF, and the elution was collected into a tube. Actin retained on the resin was eluted with 12 mL G-buffer without actin, and all elution fractions were combined and concentrated with a 30 kDa cut-off membrane (Millipore Sigma, St. Louis, MO, USA) to 0.9 mL. The 0.9 mL of each actin was polymerized by adding 100 µL of 10× MKE solution composed of 20 mM MgCl$_2$, 50 mM EGTA, and 1 M KCl for 1 hr at room temperature. The polymerized actin samples were pelleted down by ultracentrifugation at room temperature (45,000 rpm for 1 hr, TLA-55 rotor, Beckman Coulter, Indianapolis, IN, USA). The pellets were rinsed once with a 1 mL G-buffer and re-suspended into an ice-cold 0.5 mL G-buffer. The actin was depolymerized by dialysis against a 1 L G-buffer at 4°C for 2 days. The dialysis buffer was exchanged every 12 hr. The solutions with the depolymerized actin were collected into 1.5 mL centrifuge tubes and stored on ice.

## Sample preparation and cryo-EM data collection

For cryo-EM studies, G-actin was polymerized by the addition of a 10× stock solution of buffer containing 0.5 M KCl, 20 mM MgCl$_2$, 10 mM EDTA, and 0.1 M MOPS pH 7.0 before plunge freezing. Briefly, C-flat 1.2/1.3 gold grids were glow-discharged with a PELCO easiGlow (TedPella, Redding, CA, USA) for 60 s. 4 µL of actin sample was applied to the grids, incubated for 1 min, and blotted for 4 s at 95% humidity. Grids were plunged into liquid ethane using a Leica EM GM2 plunger (Leica Microsystems, Wetzlar, Germany) and stored in liquid nitrogen until data collection. All grids were screened for homogeneous sample distribution and optimal ice thickness on a Glacios cryo-TEM (Thermo Fisher Scientific, Waltham, MA, USA) equipped with a Falcon 3EC direct electron detector at a magnification of 92,000×. Data collection on optimal grids was performed on a Titan Krios G3i (Thermo Fisher Scientific, Waltham, MA, USA) operated at 300 kV, equipped with a K3 direct electron detector, a Bioquantum energy filter, and a Cs image corrector. For skeletal muscle α-actin, a total of 2046 movies with a magnification of 81,000×, corresponding to a pixel size of 0.4455 Å, were collected in super-resolution mode at a defocus range of –0.5 µm to –2.5 µm with a total electron dose of 65 e⁻/Å² per movie. For cardiac muscle α-actin, a total of 1444 movies with a magnification of 81,000×, corresponding to a pixel size of 0.4455 Å were collected in super-resolution mode at a defocus range of –0.5 µm to –2.5 µm with a total electron dose of 60 e⁻/Å² per movie. For acetylated β- and γ-actins, a total of 1352 movies and 2952 movies with a magnification of 81,000×, corresponding to a pixel size of 0.4455 Å were collected in super-resolution mode at a defocus range of –0.5 µm to –2.5 µm with a total electron dose of 50 e⁻/Å² and 65 e⁻/Å² per movie, respectively. Data collection for all actins were performed using EPU software (Thermo Fisher Scientific, Waltham, MA, USA). Data collection statistics are shown in *Table 1*.

## Image processing and 3D reconstruction

All raw movies of skeletal muscle and cardiac muscle α-actin were aligned, drift corrected, and dose weighted using the Patch motion module, and the Contrast Transfer Function (CTF) parameters were

estimated using the Patch CTF module implemented in cryoSPARCv3.2 (*Punjani et al., 2017*). All raw movies of acetylated β- and γ-actin were aligned and motion-corrected using the Patch motion module in cryoSPARC and MotionCor2 software (*Punjani et al., 2017*; *Zheng et al., 2017*), and the CTF parameters were estimated using the Patch CTF module implemented in cryoSPARCv3.2. Segments of actin filaments were initially picked using the template-free tracing method implemented in the filament tracer module in cryoSPARC to generate templates for particle picking (*Punjani et al., 2017*). A small set of particles for reference-free 2D classifications was selected and subsequently used as templates for filament tracing. Furthermore, segments of filaments were extracted, several rounds of 2D classifications were performed to remove erroneous picks, and well-aligned 2D classes were selected as templates for ab initio 3D reconstruction. For all actins, a major class with a filamentous model that showed clear helical features was selected for 3D refinements using the helical refinement module implemented in cryoSPARCv3.2 (*Punjani et al., 2017*). For acetylated β- and γ-actins, final rounds of refinements were performed using the homogenous refinement module in cryoSPARC. Data collection statistics, image processing, and the refinement summary of models are shown in *Table 1*.

## Model building, refinement, and validation

For model building, sharpened maps were used to build models of filamentous actins. We have used a common approach of model building, refinement, and validation for all actin isoforms. Briefly, the high-resolution structures of $Mg^{2+} \cdot ADP$ bound skeletal muscle and cardiac muscle α-actins, Nt-acetylated β-, and γ-actins were built using PDB entry 6DJO as a template. Initial rigid-body docking into the cryo-EM reconstructions of actin maps was performed with molecular dynamics flexible fitting function in Chimera (*Pettersen et al., 2004*). Iterative model building was performed using real-space refinement in Phenix (*Adams et al., 2010*) and COOT (*Emsley et al., 2010*). The acetylated N-terminus of the β- and γ-actins was manually placed in the cryo-EM densities using COOT. It is important to note that the N-terminal amino acids (1–3) of β- and γ-actins showed high flexibility, and we manually placed the best possible rotamers in this region by using cryo-EM map contour levels of ≥1.8–2.0σ in COOT. All rotamers were minimized and refined in Phenix. We manually built numerous parts in four actin isoform structures including the D-loop, N- and C-terminal residues, and amino acid substitutions. Furthermore, we manually modeled PTMs and placed the $Mg^{2+} \cdot ADP$ in the nucleotide-binding cleft active site and adjusted rotamers for the side chains as necessary. All final actin models were refined using Phenix, and the refined models were validated using MolProbity (*Williams et al., 2018*). Amino acids are numbered according to the sequence of mature actin isoforms (*Figure 1—figure supplement 1C*).

## Acknowledgements

We thank Dr. Jim Sellers (National Heart, Lung, and Blood Institute, National Institutes of Health) and Dr. Earl Homsher (University of California, Los Angeles) for bovine cardiac muscle α-actin powder. Electron microscopy was performed at the Center for Electron Microscopy and Analysis (CEMAS) at the Ohio State University. We thank the Ohio Supercomputer Center (OSC) for high-performance computing resources. This work was supported by the National Institute of General Medical Sciences of the National Institutes of Health under Award Numbers R01GM143539 (KC) and R01GM143414 (SMH), the National Heart Lung and Blood Institute of the National Institutes of Health under Award Number K22HL131869 (SMH), and Wellcome Trust (203276/Z/16/Z), European Research Council (ERC-2014-ADG No. 671083) and Biotechnology and Biosciences Research Council (BB/S003789/1) to MKB.

## Additional information

### Competing interests

Mohan K Balasubramanian: Reviewing editor, eLife. The other authors declare that no competing interests exist.

## Funding

| Funder | Grant reference number | Author |
| --- | --- | --- |
| National Institutes of Health | R01GM143539 | Krishna Chinthalapudi |
| Wellcome Trust | 203276/Z/16/Z | Mohan K Balasubramanian |
| European Research Council | ERC-2014-ADG No. 671083 | Mohan K Balasubramanian |
| Biotechnology and Biological Sciences Research Council | BB/S003789/1 | Mohan K Balasubramanian |
| National Institutes of Health | K22HL131869 | Sarah M Heissler |
| National Institutes of Health | R01GM143414 | Sarah M Heissler |
| Biotechnology and Biological Sciences Research Council | BB/X512023/1 | Mohan K Balasubramanian |

The funders had no role in study design, data collection and interpretation, or the decision to submit the work for publication. For the purpose of Open Access, the authors have applied a CC BY public copyright license to any Author Accepted Manuscript version arising from this submission.

## Author contributions

Amandeep S Arora, Data curation, Formal analysis, Validation, Investigation, Visualization, Methodology, Writing – review and editing; Hsiang-Ling Huang, Formal analysis, Investigation, Methodology, Writing – review and editing; Ramanpreet Singh, Yoshie Narui, Investigation, Methodology, Writing – review and editing; Andrejus Suchenko, Krishna Chinthalapudi, Conceptualization, Resources, Data curation, Formal analysis, Supervision, Funding acquisition, Validation, Investigation, Visualization, Methodology, Writing – original draft, Project administration, Writing – review and editing; Tomoyuki Hatano, Investigation, Methodology; Sarah M Heissler, Formal analysis, Funding acquisition, Validation, Investigation, Visualization, Methodology, Writing – original draft, Writing – review and editing; Mohan K Balasubramanian, Resources, Formal analysis, Supervision, Funding acquisition, Validation, Investigation, Visualization, Methodology, Writing – original draft, Writing – review and editing

### Author ORCIDs
Amandeep S Arora ⬝ http://orcid.org/0000-0003-1045-1967
Tomoyuki Hatano ⬝ http://orcid.org/0000-0002-9092-3989
Mohan K Balasubramanian ⬝ http://orcid.org/0000-0002-1292-8602
Krishna Chinthalapudi ⬝ http://orcid.org/0000-0003-3669-561X

### Decision letter and Author response
Decision letter https://doi.org/10.7554/eLife.82015.sa1
Author response https://doi.org/10.7554/eLife.82015.sa2

# Additional files

## Supplementary files
• MDAR checklist

## Data availability
All the structures and electron density maps generated have been deposited in the Protein Data Bank (PDB) and Electron Microscopy Data Bank (EMDB). The PDB and EMDB entries are 8DMX and EMD-27548; 8DMY and EMD-27549; 8DNH and EMD-27572; 8DNF and EMD-27565.

The following datasets were generated:

| Author(s) | Year | Dataset title | Dataset URL | Database and Identifier |
|---|---|---|---|---|
| Arora AS, Huang HL, Heissler SM, Chinthalapudi K | 2023 | Cryo-EM structure of skeletal muscle alpha-actin | https://www.rcsb.org/structure/8DMX | RCSB Protein Data Bank, 8DMX |
| Arora AS, Huang HL, Heissler SM, Chinthalapudi K | 2023 | Cryo-EM structure of cardiac muscle alpha-actin | https://www.rcsb.org/structure/8DMY | RCSB Protein Data Bank, 8DMY |
| Arora AS, Huang HL, Heissler SM, Chinthalapudi K | 2023 | Cryo-EM structure of nonmuscle beta-actin | https://www.rcsb.org/structure/8DNH | RCSB Protein Data Bank, 8DNH |
| Arora AS, Huang HL, Heissler SM, Chinthalapudi K | 2023 | Cryo-EM structure of nonmuscle gamma-actin | https://www.rcsb.org/structure/8DNF | RCSB Protein Data Bank, 8DNF |
| Arora AS, Huang HL, Heissler SM, Chinthalapudi K | 2023 | Cryo-EM structure of skeletal muscle alpha-actin | https://www.ebi.ac.uk/emdb/EMD-27548 | EMDB, EMD-27548 |
| Arora AS, Huang HL, Heissler SM, Chinthalapudi K | 2023 | Cryo-EM structure of cardiac muscle alpha-actin | https://www.ebi.ac.uk/emdb/EMD-27549 | EMDB, EMD-27549 |
| Arora AS, Huang HL, Heissler SM, Chinthalapudi K | 2023 | Cryo-EM structure of nonmuscle beta-actin | https://www.ebi.ac.uk/emdb/EMD-27572 | EMDB, EMD-27572 |
| Arora AS, Huang HL, Heissler SM, Chinthalapudi K | 2023 | Cryo-EM structure of nonmuscle gamma-actin | https://www.ebi.ac.uk/emdb/EMD-27565 | EMDB, EMD-27565 |

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
