## [Editor Report]

This study presents four high quality cryo-EM structures of ADP-actin filaments formed from skeletal α-, cardiac α-, cytoplasmic β- and cytoplasmic γ-actin. These structures are important for understanding the functional differences among these actin isoforms. This work is of significant general interest, because actin filaments, composed of different actin isoforms, have a critical role in a number of physiological processes from muscle contraction to cell migration and division.

---

## [Decision Letter]

**Decision letter after peer review:**

Thank you for submitting your article "Structural Mechanisms of Actin Isoforms" for consideration by *eLife*. Your article has been reviewed by 3 peer reviewers, and the evaluation has been overseen by a Reviewing Editor and Anna Akhmanova as the Senior Editor. The following individual involved in review of your submission has agreed to reveal their identity: Vitold Galkin (Reviewer #1).

Essential revisions:

1) The authors should carry out much deeper comparative analysis of the four actin filament structures at the side chain level.

2) The manuscript text should be extensively revised to remove all the misleading claims of novelty, and by re-writing or deleting the sections focusing on PTMs and analysis of actin mutations, because these do not depend on or take advantage of the new structures presented in the manuscript.

3) The figures should be extensively modified based on the reviewers' comments below.

*Reviewer #1:*

The paper by Arora et al. is aimed to structurally compare the four major actin isoforms using high-resolution cryo-EM approach. Such a comparison would be of a great interest to a broad community due to high importance of actin cytoskeletal structures for almost every cytoskeletal process from muscle contraction to cell division. The structures of the actin filaments comprised of β and γ isoforms remained unknown. Therefore, the structures obtained by the authors are important. Nevertheless, the manuscript lacks careful discussion of the obtained structures which does not allow to fully appreciate the otherwise exciting experimental data.

The following suggestions would improve the paper:

The structure of the cardiac α actin from the cardiac thin filament has been published (PDBs 6KLL and 6KLN), and this should be acknowledged. The two structures should be compared with the reconstraction of the bare cardiac F-actin.

In the Figure 1 meshwork should be replaced with the transparent surfaces to allow readers to see side chains. The N-termini densities should be magnified. Page 3, lanes 24-25 replace g- and a- with Greek letters.

Why the NTs of the β and γ actins could be resolved while the NT of the α skeletal isoform was mostly disordered and, therefore, not visible in the map?

Explanation should be provided in the light of the sequence differences or experimental procedures.

The statement about the interactions within the actin filament at the subdomain level is not adequate – the authors have excellent high resolution information to use. A detailed description of the interfaces between actin subunits within the filament for each isoform should be provided in the manuscript. Differences in the positions of amino acids involved in the filament maintenance between the isoforms (if any) should be emphasized.

Figure 2 shows significant difference in the coordination of the divalent cation and nucleotide between the isoforms. At the same time authors on page 8 state that those are identical. Is Figure 2 misleading? Show superimpositions to demonstrate the identity.

What does it mean on page 7: "Overall, the amino acid sequence divergence is higher between muscle and nonmuscle actins than within muscle or nonmuscle actin sequences"?

Discuss every non-conserved substitution in the light of its possible role in polymerization kinetics and structural difference between the isoforms. How comparison of the obtained structures explains those substitutions in the light of actin evolution?

It should be discussed why only two PTMs were selected in the discussion part. Refereeing to the published work is not enough. Methylation of the H73 is not shown.

In Figure 4 the superimposition of the β and γ actins with the actomyosin complex shows tremendous misalignment in the whole SD2 region of actin. Why the authors mention only the D-loop? How the structures were aligned, how similar were the symmetries between the structures?

The alteration of the D-loop upon myosin binding has been revealed for Myosin 15 (Gong et al., Sci Adv. 2022). This should be acknowledged in the manuscript and discussed accordingly.

The fact that the N-terminus of skeletal a-actin points in a different direction compared to those from ß-actin and g-actin does not provide any mechanistic explanation regarding the difference in the charge between the isoforms and the formation of the weakly bound complex. The authors should be more specific while discussing their hypothesis.

What about the cardiac actin and its comparison with the available cardiac actomyosin complex where loop 2 is not involved in the interactions in the rigor state? How this is related to the differences between the cytoplasmic and cardiac skeletal isoforms?

Does it mean that β and γ actins should have very different kinetics due to extra negative charge in comparison with the α actin if the same myosin isoform is used? Please, discuss it.

Particular actins work with particular myosins. For example, cardiac actin will never be used within the cell paired with cytoplasmic myosin. The authors should provide discussion how actin isoforms match to myosin motors they work together in the light of their structural data.

*Reviewer #2:*

This study presents four, high quality cryo-EM structures of ADP-actin filaments formed from four pure actin isoforms. These structures are essential for understanding the established but relatively small functional differences among these actin subunits and filaments. However, the presentation falls short of explaining these differences, the figures are difficult to appreciate, and the long sections pointing out the locations of amino acid differences and posttranslational modifications neither depend on or take advantage of the new structures. The section on myosin interactions is based entirely on previous work.

The new structural results can form the basis of an interesting and important paper, but that would depend on a new version that focuses on a much deeper structural analysis supported by much better figures.

1. Figures: The renderings of the maps and molecular models do not show the structural features of interest. For example, the space filling models in Figure 1A-D show nothing that is not already known. The same is true of panel E where the magnifications are too low to see any of the features revealed by the new work. The images of the maps with models in the third and fourth columns are so small, dense and crowed, that little can be seen. Figure 2 is meant to show the models superimposed on the maps, but maps are nearly impossible to see, and the magnifications are far too low. The diagrams in the second row are not aligned in the same way, which is confusing and sloppy.

2. Analysis of the structures: Since the four new structures compare molecules with almost identical backbone structures, the interesting structural features are only evident at the level of side chains, which are not displayed except in isolated strands in Figure 1E column 2. Do any of the sequence differences change the local structure of the filaments? Do any influence the interactions between subunits in the filaments or with the ADP? Are there compensatory substitutions locally? An appropriately deep analysis of the sequence differences is missing.

3. Amino acid substitutions: Figure 3 provides an interesting overview, but knowing the distributions of these substitutions is really only useful with a deeper analysis at the level of side chains, as noted in (3). At the overview level the absence of substitutions in SD2 is the most interesting observation. Are the residues in SD1 and SD3 that interact with the D-loop also conserved? Posing and answering this type of deeper question would make this paper much stronger.

4. Posttranslational modifications: the text emphasizes that these actin isoforms have their native posttranslational modifications, which are exactly the same ones (acetylated N-terminus and 3-methyl histidine) present in the actin subunits used for previous filament structures, so this is not novel.

5. The section "Myosin modulates actin filament structure" is largely a review of the literature that does not depend heavily on the new structures. It might be left for a review rather than being in a primary research paper.

6. Clinical mutations: the text states "our structures enabled us to reveal the location of disease-causing mutations." In fact, the new structures were not necessary to locate these sites, which were known from previous structures or could be inferred from homology models. Therefore, the section on "Amino acid substitutions overlap with the location of mutations and PTMs in actin isoforms" is largely a literature review that does not depend on the new structures. Figure 6 with very low magnification renderings is not useful without consideration of the details at the side chain level.

Suggestions and comments about the text and figures:

I would be better to write "skeletal muscle α-actin" throughout.

The text states "The gene products (for actin isoforms) are structurally and functionally conserved among eukaryotes.…" Do the authors mean "among vertebrates?" Surely not among all eukaryotes.

"Differential" in the following a poor choice of words: "isoforms display differential biochemistries.."

"We show that the defining feature used to regulate the interaction with binding proteins is the divergent N-terminus…" but the full N-termini are resolved in only two of the structures.

Page 3, lines 25-26. Symbol font missing three times.

Most investigators use nucleotide-binding cleft and barbed end groove rather than "inner and outer cleft."

Many of the fonts in Figure 1, supplement 2 are far too small to read.

Page 7: the text states "Overall, the amino acid sequence divergence is higher between muscle and nonmuscle actins than within muscle or nonmuscle actin sequences (Figure 1—figure supplement 1C)." This has been known for decades and is not new to this study.

Page 10-11: "We found that the binding of myosin to actin causes subtle conformational changes in the actin filament (Figure 4)." These conformational changes should be described and interpreted in more detail, but the whole section should probably be deleted.

Page 12: How does the "subtle inward movement of the D-loop" impact the interactions along the long pitch helix? Again, did "myosin-binding induces subtle changes in the outer actin cleft" change the separation between SD1/3 or SD2/4?

*Reviewer #3:*

In this study, the authors solve high-resolution structures of bare Mg^2+^.ADP homotypic filamentous actin using pure skeletal α-, cardiac α-, cytoplasmic β- and cytoplasmic γ-actin, including their N-termini, which have been previously elusive to structural studies. They use these data to analyze the differences in protein binding and PTMs, which likely contribute to divergent actin isoform functions in vivo.

The major strengths of the methods and results is the use of purified actin isoforms, some of which, e.g., γ cytoplasmic actin, have been previously impossible to obtain biochemically. Furthermore, this is the first resolution of the N-termini of actin protomers, which have been excluded from all the previously published structures. This is important because the N-termini harbor the major divergence between actins and thus can potentially offer key structural/functional insights into their differences in vivo.

By comparing the structures of previously published myosin-actin complexes with the bare actin structures, the authors identify subtle differences between the myosin bound and bare actin filaments, with some isoform specific differences dependent on both the myosin and the actin involved in the interaction. The authors also provide a detailed summary of where disease causing mutations in actin isoforms map on the actin structure and suggest how modifications of the physiological "actin-PTM code" could be the molecular basis of diseases caused by mutations in actin proteins.

Method-wise, the authors have achieved their goal of solving actin isoform structures. However, the novelty of the conclusions is somewhat overstated and would need to be presented more in the context of prior work in the field, which predicted the N-termini to be important in maintaining actin isoforms' divergence and their role in myosin binding, as well as some of the implications of the disease-causing mutations and PTMs.

Overall, this work will inform the field about the most likely functional determinants that drive the diversity of actin isoforms and "actin-PTM code" in vivo. The results of this study will enable future research elucidating specific actins' functions in different tissues and cell types.

In my view, experiments and data analysis are sufficient, so all the suggestions below concern changes in the text.

Overall changes:

1. It would be helpful if the authors included a more prominent discussion on the background on the actin isoforms, including the fact that the amino acid sequences of all these actins are conserved in higher vertebrates, so that there are no heterogeneities introduced into the present study by using actins from different species. It also appears important to include an upfont clarification about the source of the actin isoforms used in this study, including the fact that some of them are native and some expressed in Pichia pastoris. While these details can be gathered from the manuscript overall, I felt that stating them more clearly at the beginning could help orient the reader.

2. While the novelty of analyzing multiple actin isoforms and resolving the actins' N-termini is without question, some of the other novelty claims in this paper are overstated. Other studies mapped actin disease-causing mutations on the structure. Even more importantly, it has been long known that N-terminus is the major source of differences between actin isoforms, based on sequence alignments. The role of the N-termini in the interactions with actin-binding proteins and particularly myosin has also been proposed. The authors should word their findings in the context of these prior studies.

3. Stating that the authors analyzed the effect of Nt-acetylation of non-muscle actins on the interaction with myosin motors (Page 12, Line 12) is not quite accurate. Based on the structures of Nt-acetylated actins that the authors solved, they can speculate that the N-terminal acetyl group may contribute to the myosin loop 2 interaction with actin, but they have not "analyzed" the effect of Nt-acetylation formally. Furthermore, the actin N-terminus is highly negatively charged even without the Nt-acetylation. The authors propose that the additional negative charge introduced by the acetyl group likely enhances the long-range electrostatic interactions with the positively charged myosin residues in loop 2. This is speculative and does not appear to be strongly supported by data. This should be reflected in the text.

Specific line comments:

1. Page 3, Line 25-26: g-actin should be "γ-actin". Also, skeletal a- and cardiac a-actin should be skeletal "α- and cardiac α-actin".

2. Page 3, Lines 27-29: The authors refer to skeletal muscle actin N-terminus being unresolved and refer to Figure 1B. However, Figure 1B is labelled as cardiac actin. Also, looking at the density of the N-termini in Figures 1A-D, Figure 1B a-1 protomer seems to have the least density at what would be the N-terminus, which would make this the skeletal muscle actin? Is this a labeling error? Or is this because Figure 1B appears to be slightly rotated around the filament long axis compared to the other three structures, thus not showing the N-terminus fully at this angle in a-1 protomer. This should be corrected.

3. Page 8, Line 26: Actually, V9 is in β-actin, while in γ-actin it is I9. While this is a conservative substitution, it would be interesting to know if this has any effect on the nucleotide binding and/or dynamic properties, or interaction with some regulatory proteins. The authors should comment on this, if possible.

4. Figure 4: The authors refer to the rigor state as apo in the text. Either the rigor nomenclature or the apo nomenclature should be followed for consistency.

5. Page 12, Line 9: I believe the authors are referring to Figure 4B, not Figure 5B.

6. Page 12, Line 15 and Figure 5: The authors refer to Figure 5B, but I believe they are actually talking about Figure 5A. Figure 5A labeling needs to be clearer. I believe the teal and yellow labels are referring to Myosin-1B bound α-actin, and the magenta to NM2C bound γ-actin, while the blue and dark red are bare cytoplasmic actins. Instead of labeling the structures with just the motor protein, the authors need to label which actin isoform is bound to the labeled motor. It might help if the other colors in this figure are suffused, so that the ones emphasized stand out more.

7. Page 12, Line 26: The authors talk about the shorter loop-2 in M1B being less efficient at stabilizing the actin N-terminus. It is unclear if they are referring to how the α-actin N-terminus points in a different direction in M1B bound structures compared to NM2C bound γ-actin N-terminus. If so, why is this referred to as stabilizing the N-terminus?

[Editors' note: further revisions were suggested prior to acceptance, as described below.]

Thank you for resubmitting your work entitled "Structural Mechanisms of Actin Isoforms" for further consideration by *eLife*. Your revised article has been evaluated by Anna Akhmanova (Senior Editor) , a Reviewing Editor, and one of the original reviewers.

The manuscript has been improved but there are important remaining issues that need to be addressed, as outlined below:

Although both manuscript text and figures have been modified in the revised version, there are still several issues that should be addressed before the manuscript is acceptable for publication. Reviewer #2 has, therefore, provided a detailed list of revisions (see below). Thus, both the manuscript text and several figures should be extensively revised to improve the data presentation, as well as the focus of the manuscript.

*Reviewer #2:*

The new structures seem to be well done and are a valuable contribution. The authors made many changes in response to the first round of reviews. Unfortunately, these changes did not improve the paper substantially. The authors revised the figures, but the new figures have the same problems as the originals. The text of the article has much new material (in response to the reviews) but the result is that the presentation is now poorly organized, and still contains unwarranted or inaccurate statements.

Making acceptable figures and rewriting the text should be straightforward but so far has escaped the authors. I would like to help but cannot rewrite the paper. The following is an incomplete list of issues that need to be addressed:

• Title: Not appropriate. The paper mostly describes some new structures. The only mechanistic advance is the first views of the complete N-termini of two actins with some ideas about how they might influence interactions with myosin.

• Abstract: The focus is wrong; it contains nothing specific about how the new structures are the same or different from each other. What is a "retropropagated structural change?"

• Introduction: The text attributes to differences in actin isoforms "the formation of diverse cellular actin networks with distinct compositions, architectures, dynamics, and mechanics that enable fundamental cell functions including adhesion, migration, and contractility." Small differences in the isoforms may facilitate the formation of different structures but are unlikely to be causative as claimed. Actin binding and signaling proteins surely are the dominant factors.

Results

• Maps and models: The text states "Our cryo-EM maps allowed us to build unambiguous models of actin isoforms in which secondary structure information including the side chains, the nucleotide and associated cation (Mg^2+^·ADP), and PTMs were apparent from the densities (Figure 1E, Figure 1 – —figure supplement 2)." The models were not built de novo from the maps. Rather a previous structure was docked into the maps followed by automated refinement procedures. I expect that the backbones changed very little during refinement, but this is not stated. It would be interesting to know how much refinement adjusted the side chains and handled the substitutions.

The figures do not allow a reader to evaluate the quality of the maps and models, since the maps are shown at very low contour levels, into which almost any side chain would fit. Furthermore, some of the new surface representations of the maps (Figure 1E, Figure 1S1B, Figure 2) are shown with low contrast and almost no depth cues making it difficult to see how the models fit into the maps even in the videos. Other renderings (Figure 1S3) are too dark to see details.

The models are rendered by an unspecified method with a continuous backbone that differs in thickness between figures and seems to be some kind of smoothened average rather than representing the atoms and bonds. The stick figures of side chains are better. Unfortunately, without stronger depth cues or thinner Z-slices, most of the details are lost in a confusing maze of overlapping details. To make matters worse, some points of view are not helpful: for example, viewing a histidine ring on edge is the worst way to see a methyl group. Similarly, the views of the active site are not as clear is previous papers from other labs.

The four new videos of the maps, backbone models and side chain models offer some help, but they have problems with low contrast maps and overlapping structures as noted above. Even at half speed the images rotate too quickly to see details such as the N-termini.

The deep colors used in Figure 1A-D do not show the subunits clearly. A color tinted version of Figure 1 – supplement 1 would be much better.

The font size is too small in Figure 1S2: Here are the standards for font size in figures (from JBC): Use a sans serif font such as Arial or Helvetica. Use regular font style, not bold. Use appropriately sized numbers, letters, and symbols so they are no smaller than 2 mm after reduction to a 1, 1.5 or 2 column width. Superscript and subscript characters are not excluded from this rule. Numbers, letters and symbols used in multi-paneled figures must be consistent, that is all the same font and size.

• Comparison of the structures. The organization of this section needs to be rethought. Here is a possible outline:

1. Comparison of helical parameters and subunit backbones.

Lines 130-131 belong at the beginning of the structure comparisons "No significant changes (this should be differences, since the isoforms do not interchange) in the pitch of the actin helix were observed between our structures of actin isoforms, emphasizing their overall conserved filamentous structure in the absence of myosin motors or ABPs."

Rather than being tacked on as an afterthought at the end of a paragraph about the maps, the following should be the topic sentence for a paragraph in this section: "The superposition of all actin isoform structures shows a root-mean-square deviation (RMSD) between 0.83Å to 1.04Å, indicating an overall similar topology." Are the RMSD's for all atoms or more appropriately the backbones? I expect that within experimental error the backbones are identical (rather than "overall similar").

2. Comparison of side chains. Lines 97-113 are common knowledge, so they can be deleted unless used to make specific points of comparison between the structures.

Figure 5 showing the locations of the substitutions is excellent. However, identifying these locations (lines 116-127) does not depend on the new structures. Instead, the new structures allow for the first examination the effects of substitutions on local structures (as requested in the first review). Consider are the following questions. Are the side chain rotomers identical in the isoforms or do they differ? Do the orientations of the side chains differ where there are substitutions? As requested in the first review (but not provided), are there compensatory substitutions at other positions when alternative side chains have different volumes?

Line 155: It is not true that "side chain of L16 in muscle actins" is larger than methionine. Both have 4 heavy atoms.

3. Comparison of subunit interactions in filaments.

Lines 163-177: The key findings are buried in the paragraph. (Finding 1) The transverse interprotomer interfaces (also called the short pitch helix interfaces) are identical in the actin isoforms (cite Figure 4 as the evidence and make it clear in the legend that these interfaces are identical). (Finding 2) Interactions of the D-loop in SD2 with SD1 and SD4 of the neighboring subunit are identical in the actin isoforms. If (2) is correct, how does it relate the statement "Amino acid substitutions at the longitudinal interprotomer interface include T200/V201, C271/A272, and V286/I287." Are these strictly along the long-pitch helix? What are the partner residues contacted in the adjacent subunit? Where are these differences and which interactions are affected? The rest of the material in this paragraph was already known and does not need to be repeated. Note that this study did not reveal "that longitudinal interactions are mainly mediated by hydrophobic amino acids."

I do not understand what Figure 3 supplement 1 or Figure 4 supplement 1 add to the paper if these interfaces are identical in the various isoforms.

I do not understand the legend of Figure 5. This looks like the longitudinal interface not the transverse interface, which is shown in Figure 4. In any case the legend should make clear how these subunits are oriented.

I do not understand the legend of Figure 6, which says this shows "The actomyosin interface of the native cardiac thin filament." If myosin is bound, it is not a native thin filament.

• Posttranslational modifications.

Lines 179-188 are not results from this study and can be deleted.

Lines 189-205 are not really about PTMs, rather they are about observing density for previously unobserved residues at the N-termini. I do not agree with the statement "quality of our density maps allowed us to resolve both key PTMs." The maps shown in Figure 1 – supplement 3 are not good enough to identify the methyl group on histidine. As noted above, the manuscript does not include a high magnification illustration of the maps and models for the four N-termini, so one cannot evaluate whether the acetyl group is present.

• Myosin interactions. This is the most important section of the paper. It will be better with a more concise presentation.

Line 223: Start this section with the excellent topic sentence "Myosin binding to actin filaments causes subtle conformational changes in the filament." Following this topic sentence, add lines 224-227, although local changes are more important than the overall RMSDs. Rather than listing the structures on lines 210-222, comment on what you learned from each later in the section. Finish the paragraph with lines 232-243 starting with "SD2 adopts a different conformation…."

Continue with "The superimposition of bare… " on line 248.

As a general matter, phrases such as "First, we compared," "We found that…," "Next, we compared the position…" and "Our structural comparison also revealed…" are not necessary. Just state the result.

Figure 6 supplement 1 shows the relation of the actin N-terminus to myosin particularly well, although this ribbon diagrams would be better with a wider range of colors for the actins and myosin. This view might be combined in some way with Figure 7 and shown in the main text rather than Figure 6S1. Showing the map densities for these actin N-termini is essential to convince readers that their conformations differ. How does one know if the conformations in bare filaments are the same with myosin bound?

Line 312: topology is used incorrectly.

• Discussion. Again, a more concise presentation would be better. You can go straight to the point that the structures of the subunits, filaments and interfaces between subunits in filaments are virtually identical, is spite of modest numbers of amino acid substitutions. The new structures reveal some or all of the N-termini of these actins, which seem to interact differently with myosins bound to the filaments. (Contrary to the conclusion at the end of discussion, the paper has no evidence about how the interactions of the isoforms with ABPs might differ.)

Delete lines 281-287 unless the visualization of these PTMs can be made more convincing.

---

## [Author Response]

Essential revisions:1) The authors should carry out much deeper comparative analysis of the four actin filament structures at the side chain level.2) The manuscript text should be extensively revised to remove all the misleading claims of novelty, and by re-writing or deleting the sections focusing on PTMs and analysis of actin mutations, because these do not depend on or take advantage of the new structures presented in the manuscript.3) The figures should be extensively modified based on the reviewers' comments below.

We thank the reviewers for their valuable criticism, suggestions, and positive feedback. We have extensively revised our manuscript in response to their criticism and feel that the changes have significantly improved the strength of our work. We responded to all criticisms in a point-by-point response below. The revised manuscript includes a comparative analysis of the four actin isoform structures that includes the description of longitudinal and transverse interprotomer interfaces at the side chain level. We have extensively revised figures and include two new main figures (Figures 4, 5), four new figure supplements (Figure 1 —figure supplement 1, Figure 2 —figure supplement 1, Figure 4 —figure supplement 1, Figure 6 —figure supplement 1) and 6 videos (Figure 1 – videos 1-4, Figure 6 – video 1, Figure 7 – video 1) for clear presentation of the observations and novelty in the revised manuscript. In addition, we have revised the manuscript text to remove unintentional claims of novelty and included a more extensive discussion of our findings in the context of prior studies. As suggested, we have removed the sections that focus on actin mutations and PTMs that are not resolved in our structures in the revised manuscript. We hope that our study will now be accepted for publication in *eLife*.

Reviewer #1:The paper by Arora et al. is aimed to structurally compare the four major actin isoforms using high-resolution cryo-EM approach. Such a comparison would be of a great interest to a broad community due to high importance of actin cytoskeletal structures for almost every cytoskeletal process from muscle contraction to cell division. The structures of the actin filaments comprised of β and γ isoforms remained unknown. Therefore, the structures obtained by the authors are important. Nevertheless, the manuscript lacks careful discussion of the obtained structures which does not allow to fully appreciate the otherwise exciting experimental data.The following suggestions would improve the paper:The structure of the cardiac α actin from the cardiac thin filament has been published (PDBs 6KLL and 6KLN), and this should be acknowledged. The two structures should be compared with the reconstraction of the bare cardiac F-actin.

As suggested by the reviewer, we compared the structures of the native cardiac thin filament (PDB entries 6KLL and 6KLN) with our structure of bare cardiac muscle α-actin. We noticed that the structure of cardiac muscle α-actin from the cardiac thin filament was deposited as “skeletal α-actin” and the respective PDB files correspond to the sequence of skeletal muscle α-actin. Due to this discrepancy, we decided not to include the comparison of the respective structures with our structure of bare actin in the revised manuscript. However, we now included a citation to acknowledge the work of Oda *et al.*, J Struct Biol, 2020 in the “introduction” of the revised manuscript.

Other published structures of cardiac muscle α-actin complexes including PDB entries 7LRG and 5NOL are with ~6-8 Å resolution not suitable for a comparative structural analysis – especially at the level of side chains. Instead, we now include a comparison of key regions of bare cardiac muscle α-actin with the cardiac actomyosin complex (PDB 7JH7) in Figure 6 —figure supplement 1 as suggested by this reviewer in another point of critique.

In the Figure 1 meshwork should be replaced with the transparent surfaces to allow readers to see side chains. The N-termini densities should be magnified.

We thank the reviewer for the suggestions. We have replaced the meshwork with a transparent surface representation in Figure 1E. Additionally, we included videos for actin isoform protomers (Figure 1 – videos 1-4) that show the quality of our electron density maps and resolved structural features (without and with side chains) including the N-terminus for better visualization.

Page 3, lanes 24-25 replace g- and a- with Greek letters.

We thank the reviewer for catching these typographical errors. It appears that some Greek characters were inadvertently converted into text. We have corrected all instances in the revised manuscript.

Why the NTs of the β and γ actins could be resolved while the NT of the α skeletal isoform was mostly disordered and, therefore, not visible in the map?Explanation should be provided in the light of the sequence differences or experimental procedures.

This is an interesting question. The experimental procedures were the same for all samples and unlikely to contribute to the observed differences in the number of resolved amino acids at the very N-terminus of our structures of actin isoforms. While we cannot exclude that sequence differences may have contributed, it is likely that the source of the different actin isoforms that we used in this work is the main contributor. We have recombinantly produced β- and γ-actin in *Pichia pastoris* together with N-acetyl transferase NAA80 and histidine methyl transferase SETD3 to obtain actin isoforms with uniform and complete Nt-acetylation and methylation of H72/73 for structural studies (Hatano *et al.*, J Cell Sci., 2020). By contrast, muscle actin isoforms were extracted and purified from rabbit skeletal muscle and bovine cardiac muscle and likely have a non-uniform PTM pattern with >90% Nt-acetylation (Chen *et al.*, bioRxiv, 2021), H72/73 methylation and likely other modifications (Terman and Kashina, Curr Opin Cell Biol, 2013). This non-uniform PTM pattern may have contributed to the lack of observable densities for the very first amino acid of cardiac muscle α-actin and the first three amino acids of skeletal muscle α- actin. The following sentences have been included in the “results” section of the revised manuscript:

“The lack of resolvable density for the very first amino acid of cardiac muscle α-actin and the first three amino acids of skeletal muscle α-actin may be attributed to a non-uniform PTM pattern of actin isoforms prepared from the native source (skeletal or cardiac muscle) compared to our recombinant nonmuscle actin isoforms with uniform PTM pattern in that the entire N-terminus region could be resolved (Figure 1B, Videos 1-4).”

“Like in previous cryo-EM structures of muscle α-actins, there are no resolvable densities for the very N-terminus, including Nt-acetylation, in our density maps of skeletal muscle α-actin and cardiac muscle α-actin (Figure 1E), possibly due to non-uniform PTM patterns of native, tissue- purified muscle actins.”

The statement about the interactions within the actin filament at the subdomain level is not adequate – the authors have excellent high resolution information to use. A detailed description of the interfaces between actin subunits within the filament for each isoform should be provided in the manuscript. Differences in the positions of amino acids involved in the filament maintenance between the isoforms (if any) should be emphasized.

We thank the reviewer for this excellent suggestion. We now include two new figures and one supplemental figure (Figure 4, Figure 5, Figure 4 —figure supplement 1) to show the longitudinal and transverse interprotomer interface contacts at the side chain level. The following text has been added to the “results” and “discussion” of the revised manuscript.

“Next, we analyzed longitudinal and transverse interactions at interprotomer interfaces in our four structures of actin isoforms (Figure 4, Figure 5, Figure 4 —figure supplement 1). We found that longitudinal interactions are mainly mediated by hydrophobic amino acids that are likely to enable interactions with water molecules that were recently shown to mediate interprotomer contacts within the filament core (Figure 4, Figure 4 —figure supplement 1) (55). Amino acid substitutions at the longitudinal interprotomer interface include T200/V201, C271/A272, and V286/I287. These substitutions may influence the stability of the promoters based on their ability to interact with solvent (55). The transverse interprotomer interface is formed through hydrophilic and hydrophobic interactions (Figure 5, Figure 4 —figure supplement 1). In contrast to interactions at the longitudinal interface, transverse interprotomer interactions are direct and not mediated by solvent. Notably, no amino acid substitutions between actin isoforms are present at the transverse interprotomer interface. Interactions with the D-loop located in SD2 are conserved in our structures of actin isoforms, underlining the critical and conserved role of the D-loop to mediate interprotomer interactions. At the intersection of the longitudinal and transverse interprotomer interfaces, H39/40 and H172/H173 together with M43/M44 act as central anchors.”

“The comparison of longitudinal and transverse interprotomer interfaces revealed that the transverse interface is highly conserved compared to the longitudinal interprotomer interface that features several amino acid substitutions. Further, our analysis showed the presence of conserved amino acids (H39/H40, H172/H173, M43/M44) at the intersection of the longitudinal and transverse interprotomer interfaces. These residues were previously shown to be highly susceptible to oxidative stress caused by reactive oxygen species (ROS) and suggest a possible implication for the filament stability of actin isoforms (11). Further, the absence of amino acid substitutions in SD2 and conserved interactions between the SD2 D-loop and amino acids of SD1 and SD3 suggests selective pressure to maintain this critical structural element.”

Figure 2 shows significant difference in the coordination of the divalent cation and nucleotide between the isoforms. At the same time authors on page 8 state that those are identical. Is Figure 2 misleading? Show superimpositions to demonstrate the identity.

We have updated panels A to C of *Figure 2* and now also include superimposition of the active sites and the bound nucleotide and Mg^2+^ in panels in a new figure supplement (*Figure 2 —figure supplement 1*). Comparative structural analysis shows small rearrangements due to rotamer orientations in the active site, the coordination of the divalent cation, and the nucleotide. New text has been included in the “results” section of the revised manuscript:

“The superimposition of the nucleotide cleft active sites of our four structures of actin isoforms shows small changes in the positions of the nucleotide (RMSD ~0.44-0.47 Å) and the bound Mg^2+^ (RMSD ~0.5-1.3 Å), especially in the position of the β-phosphate group relative to the β-phosphate group (Figure 2, Figure 2 —figure supplement 1). The position of the Mg^2+^ moves relative to the position of the β-phosphate group (Figure 2 —figure supplement 1). The superimposition of the nucleotide-binding cleft active sites further shows small local rearrangements of side chains of conserved amino acids (Figure 2 —figure supplement 1) including R182/R183 and K335/K336. Amino acids M15/L16 and M304/M305 form hydrophobic interactions in the nucleotide-binding cleft active site (Figure 2).”

What does it mean on page 7: "Overall, the amino acid sequence divergence is higher between muscle and nonmuscle actins than within muscle or nonmuscle actin sequences"?

We apologize for this ill-worded sentence. It has been replaced with the following sentence for better readability: “Overall, the amino acid sequence is more conserved among muscle actins than between muscle and nonmuscle actins.”

Discuss every non-conserved substitution in the light of its possible role in polymerization kinetics and structural difference between the isoforms. How comparison of the obtained structures explains those substitutions in the light of actin evolution?

While it is tempting to speculate the possible roles of individual non-conserved amino acid substitutions on polymerization kinetics of actin isoforms and the structural and functional differences in the context of actin evolution, faithful predictions are difficult to make and would need to be experimentally verified, as it has been done in a recent elegant study (Boiero Sanders, EMBO J, 2022), which is beyond the scope of the present work.

It should be discussed why only two PTMs were selected in the discussion part. Refereeing to the published work is not enough. Methylation of the H73 is not shown.

PTM patterns fluctuate in time and space and can be added to a protein during or after translation. Further, it has been shown that PTMs at the protein termini are often more abundant than modifications at other sites (Chen and Kashina, Front Cell Dev Biol., 2021). For vertebrate actins, it has been shown that the N-terminus is normally acetylated and that H72/73 is normally methylated (Drazic et al., Proc Natl Acad Sci USA, 2018; Arnesen, PLoS Biol, 2011; Kwiatkowski *et al.*, *ELife*, 2018; Johnson *et al.*, Biochem J, 1967). Both PTMs show strong evolutionary conservation. Nt-acetylation has been shown to be critical for bona fide actin structure and function and the interaction with some ABPs, while H72/73 methylation plays a role in the ATP hydrolysis pathway where it is thought to slow the release of π (Drazic et al., Proc Natl Acad Sci USA, 2018; Kabsch *et al.*, Nature, 1990; Nyman *et al.*, J Mol Biol, 2002; Terman and Kashina, Curr Opin Cell Biol, 2013, Cook et al., J Biol Chem 1992). Therefore, we chose to focus our analysis on these two PTMs in our work. To mimic this predominant PTM pattern of actin isoforms, we recombinantly produced human β- and γ-γ actin with the Pick-ya actin method that we previously developed (Hatano *et al.*, J Cell Sci., 2020). By utilizing an engineered *Pichia pastoris* strain that expresses the human N-acetyl transferase NAA80 and histidine methyl transferase SETD3, we could produce and purify actin isoforms with uniform Nt-acetylation and methylation of H73 for structural studies. Other PTMs are not detectable in these recombinant actins (Hatano *et al.*, J Cell Sci., 2020). Respective PTM patterns have been described for muscle actins while it is likely that our preparations from native sources contain additional, low-abundance PTMs that could not be resolved in our structural studies. The following statement has been included in the “results” section of the revised manuscript.

“The ‘PTM code’ is likely to change in time and space, between actin isoforms and within actin molecules and only a few PTMs have been characterized in detail (11, 54). Here, we focused our structural analysis on two widely documented and highly conserved PTMs of mature vertebrate actins: the posttranslational acetylation of the N-terminus by N-terminal acetyltransferase NAA80 and methylation of H72/H73 by SET domain protein 3 (SETD3) (57, 58). Nt-acetylation has been shown to be critical for bona fide actin structure and function and the interaction with some ABPs, while H72/73 methylation plays a role in the ATP hydrolysis pathway where it is thought to slow the release of π (25, 54, 57, 59). Both PTMs are present in actin prepared from vertebrate tissues and our preparations of recombinant human β- and γ-actin produced in an engineered Pichia pastoris strain (42). The quality of our density maps allowed us to resolve both key PTMs.” (Please add may be one sentence about the function of the modifications based on phenotypes of NAA80 and SETD3).

We now also include a new supplementary figure (*Figure 1 —figure supplement 3*) that shows methylation of H72/73 in all actin isoforms.

The discussion on additional PTMs and disease-causing mutations in actin isoforms has been deleted as suggested in the editor’s summary.

In Figure 4 the superimposition of the β and γ actins with the actomyosin complex shows tremendous misalignment in the whole SD2 region of actin. Why the authors mention only the D-loop? How the structures were aligned, how similar were the symmetries between the structures?The alteration of the D-loop upon myosin binding has been revealed for Myosin 15 (Gong et al., Sci Adv. 2022). This should be acknowledged in the manuscript and discussed accordingly.

We apologize for not discussing the movement of SD2 in more detail in the original version of the manuscript. We originally focused our analysis on the D-loop because of its key role in longitudinal and transverse interactions in the actin filament and its conformational changes upon myosin binding. We now include a new supplementary figure (Figure 6 —figure supplement 1) that compares the conformation of the D-loop and the actin N-terminus in bare cardiac muscle α-actin with those found in the structure of the cardiac thin filament/cardiac myosin-2 complex (Risi et al., Structure, 2021) We also include new videos (Figure 6 – video 1, Figure 7 – video 1) that show the structural rearrangements of the SD2 region and the N-terminus in bare and myosin-bound actin. This structural comparison shows that the SD2 is not misaligned but rather adopts a different conformation in myosin-bound compared to the bare actin structures. The following text has been included in the “results” section of the revised manuscript.

“Structural comparison of our structures of bare actin isoforms and previous structures of myosin- bound actins shows that SD2 adopts a different conformation (Figure 6A, B, Figure 6 —figure supplement 1). The conformation of the D-loop, located in SD2, differs with an RMSD of ~0.95-1.25Å (Figure 6 A,B). A direct comparison of bare g-actin with g-actin/NM2C complex shows the subtle inward movement of the D-loop with an RMSD of ~1.25Å in the myosin-bound state, suggesting that myosin-binding induces subtle changes in the barbed end groove. Notably, the inward movement of the D-loop (RMSD ~ 0.91 Å) is also observed in the recent structure of cardiac myosin bound to the thin filament compared to our structure of bare cardiac muscle α- actin (Figure 6 —figure supplement 1), suggesting that myosin and not the regulatory proteins tropomyosin and troponin drive this structural change.”

We also reference and discuss the conformation of the D-loop of the recent structure of the skeletal muscle α-actin/myosin-15 complex (Gong *et al.,* Sci Adv, 2022) in the “discussion” of the revised manuscript.

“By contrast, the recent high-resolution structures of rigor and Mg^2+^·ADP.myosin-15 bound to skeletal muscle α-actin suggests flexibility in the D-loop in the rigor state and a rigid D-loop in close proximity to the actin surface in the Mg^2+^·ADP state. The flexible D-loop has been proposed to support actin nucleation activity whereas the rigid D-loop has been proposed to limit the actin nucleation activity of this myosin (41).”

The fact that the N-terminus of skeletal a-actin points in a different direction compared to those from ß-actin and g-actin does not provide any mechanistic explanation regarding the difference in the charge between the isoforms and the formation of the weakly bound complex. The authors should be more specific while discussing their hypothesis.

In response to this reviewer and reviewer #3, we have included the following sentence in the “discussion” of the revised manuscript.

“Based on our structural and previous biochemical studies, we speculate that the biochemical properties of the very N-terminus of actin isoforms are key contributors to the formation of isoform-specific interfaces with myosin motors (16, 56, 71). For instance, the very N-termini of mature β- actin (DDD) and γ-actin (EEE) have different pKa values, resulting in a higher negative electron charge density in β-actin compared to γ-actin that is further increased by Nt-acetylation in both isoforms. These differences are likely to contribute to distinct interactions with the positively charged loop-2 and other actin-binding elements in the motor domain of myosin family members during the formation of the actomyosin interface.”

Also, please see our responses to the specific questions raised by this reviewer in another point of critique and reviewer #3.

What about the cardiac actin and its comparison with the available cardiac actomyosin complex where loop 2 is not involved in the interactions in the rigor state? How this is related to the differences between the cytoplasmic and cardiac skeletal isoforms?

We are surprised by the statement that the cardiac myosin loop-2 is not involved in interactions with the actin filament in the rigor state. To the best of our knowledge, interactions between loop- 2 and F-actin have been extensively characterized across the myosin superfamily (May be only mention below the reference that relates to cardiac actin-myosin-II loop2 interactions)(Uyeda *et al.*, Nature, 1994; Furch *et al.*, Biochemistry, 1998; Yengo *et al.*, Biochemistry 2004; Joel *et al.*, J Biol Chem, 2001; Onishi *et al.*, Proc. Natl. Acad. Sci. USA, 2006; Struchholz *et al.*, J. Biol. Chem. 2009) and have been reported in all cryo-EM structures of actomyosin complexes where this loop has been resolved/partially resolved (von der Ecken *et al.*, Nature, 2016; Gurel *et al.*, *eLife*, 2017; Mentes *et al.*, Proc Natl Acad Sci USA, 2018) including the recent structure of the cardiac actomyosin complex (Risi *et al.*, Structure, 2021). While the tip of loop-2 is not resolved in this structure, its base is resolved and forms a hydrogen bond with actin as reported in Risi *et al.* Structure 2021.

Does it mean that β and γ actins should have very different kinetics due to extra negative charge in comparison with the α actin if the same myosin isoform is used? Please, discuss it.Particular actins work with particular myosins. For example, cardiac actin will never be used within the cell paired with cytoplasmic myosin. The authors should provide discussion how actin isoforms match to myosin motors they work together in the light of their structural data.

We thank the reviewer for this comment and the suggestion. We previously reported key kinetic and functional parameters of human nonmuscle myosin-2A, -2B, and -2C as well as human myosin-7A assayed with skeletal muscle α-actin, β-actin, and γ-actin (Muller *et al.*, PLoS One, 2013). These experiments revealed kinetic and functional differences in the interaction between actin isoforms and individual myosin motors. We would also like to highlight that “cytoplasmic” or “nonsarcomeric” myosins play key roles in adult muscle and during muscle development (Hammers *et al.*, *eLife*, 2021; Fenix *et al.*, *eLife*, 2018; Horsthemke *et al.*, J Biol Chem, 2019; Karolczak *et al.*, Histochem Cell Bio, 2013), suggesting that they may indeed interact with muscle actins in vivo. The following sentence has been included in the “discussion” section of the revised manuscript:

“For example, previous in vitro experiments with recombinant proteins revealed isoform-specific differences in the interaction between muscle and nonmuscle actin isoforms and individual myosin motors (16). Notably, predominantly nonsarcomeric myosins have a faster ATPase activity and in vitro sliding velocity when assayed with nonmuscle actins compared to muscle actin, suggesting selective fine-tuning of the functional competencies of myosin-actin combinations with implications for cell function (16). These in vitro observations are supported by our structural data that suggest that the different orientations of the N-termini of actin isoforms contribute to the formation of different binding interfaces with myosin motors (Figure 7). Previous work also revealed differences in the extent to which the highly conserved β-actin and γ-actin isoforms (*Figure 1—figure supplement 1*) can activate the kinetic and functional activity of the same motor in vitro (16). Based on our structural and previous biochemical studies, we speculate that the biochemical properties of the very N-terminus of actin isoforms are key contributors to the formation of isoform-specific interfaces with myosin motors (16, 56, 71). For instance, the very N- termini of mature β-actin (DDD) and γ-actin (EEE) have different pKa values, resulting in a higher negative electron charge density in β-actin compared to γ-actin that is further increased by Nt- acetylation in both isoforms. These differences are likely to contribute to distinct interactions with the positively charged loop-2 and other actin-binding elements in the motor domain of myosin family members during the formation of the actomyosin interface.”

We have also included a statement about the expression of multiple actin isoforms in most cells in the “introduction” and a statement about the mixing and matching of actin isoforms and myosin motors in the “discussion”

“Most cells maintain a defined ratio of actin isoforms with muscle and nonmuscle actins representing the main isoforms in muscle and non-muscle cells, respectively (3-8). Actin isoforms have specific and redundant roles in cells and display different biochemistries, cellular localization, and interactions with myosin motors and actin-binding proteins (ABPs) (2, 3, 9-19). This cell type-specific mixing and matching is likely to fine-tune the enzymatic output of myosin motors for specialized functions (6, 16, 78).”

Reviewer #2:This study presents four, high quality cryo-EM structures of ADP-actin filaments formed from four pure actin isoforms. These structures are essential for understanding the established but relatively small functional differences among these actin subunits and filaments. However, the presentation falls short of explaining these differences, the figures are difficult to appreciate, and the long sections pointing out the locations of amino acid differences and posttranslational modifications neither depend on or take advantage of the new structures. The section on myosin interactions is based entirely on previous work.The new structural results can form the basis of an interesting and important paper, but that would depend on a new version that focuses on a much deeper structural analysis supported by much better figures.1. Figures: The renderings of the maps and molecular models do not show the structural features of interest. For example, the space filling models in Figure 1A-D show nothing that is not already known. The same is true of panel E where the magnifications are too low to see any of the features revealed by the new work. The images of the maps with models in the third and fourth columns are so small, dense and crowed, that little can be seen. Figure 2 is meant to show the models superimposed on the maps, but maps are nearly impossible to see, and the magnifications are far too low. The diagrams in the second row are not aligned in the same way, which is confusing and sloppy.

We agree with the reviewer that the structure and functions of filamentous actin have been extensively studied for decades (Egelman et al., Nature 1982; Holmes et al., Nature, 1990; Chou and Pollard, Proc Natl Acad Sci USA, 2019; Belyy et al., PLoS Biol, 2020; Merino et al., Nat Struct Mol Biol, 2018; Mei et al., *ELife*, 2020; Mentes et al., Proc Natl Acad Sci USA, 2018; Oda et al., Nature, 2009). While the significant structural differences that we observed in our structures of four actin isoforms are not highlighted in the space-filling models shown in Figure 1A-D, we believe that it is very important to show the overviews of the actual experimental data not only to highlight the structural conservation but also to serve as a point of reference for the non-expert reader. Figure 1E has been modified in response to a critique raised by reviewer #1. The meshwork representation has been replaced with a transparent surface representation and four videos (Figure 1 – videos 1-4) have been included that show the quality of our electron density maps and resolved structural features in the actin isoform protomer for better visualization.

We have updated all panels in Figure 2 and now additionally show a superimposition of the four active sites and the bound Mg^2+^ and ADP in Figure 2 —figure supplement 2.

2. Analysis of the structures: Since the four new structures compare molecules with almost identical backbone structures, the interesting structural features are only evident at the level of side chains, which are not displayed except in isolated strands in Figure 1E column 2. Do any of the sequence differences change the local structure of the filaments? Do any influence the interactions between subunits in the filaments or with the ADP? Are there compensatory substitutions locally? An appropriately deep analysis of the sequence differences is missing.

We agree with the reviewer that a more detailed analysis of the side chains and local structural changes strengthens our work. Since figures tend to get too busy when densities are shown in addition to the model, we included four videos (*Figure 1 – videos 1-4*) that show the overall quality of the maps and structures of the four actin isoforms. The specific questions of the reviewer were addressed as follows:

1. Do any of the sequence differences change the local structure of the filaments?

Please see our response to question 2.

2. Do any influence the interactions between subunits in the filaments or with the ADP?

We included a comparative structural analysis of interactions along the longitudinal and transverse interprotomer interfaces (*Figure 4,5, Figure 4 —figure supplement 1*) of all actin isoforms in the revised manuscript. We also performed a more detailed analysis of the active site and the positions of the nucleotide and Mg^2+^ in our structures of actin isoforms (*Figure 2 —figure supplement 1*). The following sentences have been included in the revised manuscript.

“Next, we analyzed longitudinal and transverse interactions at interprotomer interfaces in our four structures of actin isoforms (Figure 4, Figure 5, Figure 4 —figure supplement 1). We found that longitudinal interactions are mainly mediated by hydrophobic amino acids that are likely to enable interactions with water molecules that were recently shown to mediate interprotomer contacts within the filament core (Figure 4, Figure 4 —figure supplement 1) (55). Amino acid substitutions at the longitudinal interprotomer interface include T200/V201, C271/A272, and V286/I287. These substitutions may influence the stability of the promoters based on their ability to interact with the solvent (55). The transverse interprotomer interface is formed through hydrophilic and hydrophobic interactions (Figure 5, Figure 4 —figure supplement 1). In contrast to interactions at the longitudinal interface, most of the transverse interprotomer interactions are direct and not mediated by solvent. Notably, no amino acid substitutions between actin isoforms are present at the transverse interprotomer interface. Interactions with the D-loop located in SD2 are conserved in our structures of actin isoforms, underlining the critical and conserved role of the D-loop to mediate interprotomer interactions. At the intersection of the longitudinal and transverse interprotomer interfaces, H39/40 and H172/H173 together with M43/M44 act as central anchors.”

“The superimposition of the nucleotide cleft active sites of our four structures of actin isoforms shows small changes in the positions of the nucleotide (RMSD ~0.44-0.47 Å) and the bound Mg^2+^ (RMSD ~0.5-1.3 Å), especially in the position of the β-phosphate group relative to the β-phosphate group (Figure 2, Figure 2 —figure supplement 1). The position of the Mg^2+^ moves relative to the position of the β-phosphate group (Figure 2 —figure supplement 1). The superimposition of the nucleotide-binding cleft active sites further shows small local rearrangements of side chains of conserved amino acids (Figure 2 —figure supplement 1) including R182/R183 and K335/K336. Amino acids M15/L16 and M304/M305 form hydrophobic interactions in the nucleotide-binding cleft active site (Figure 2). Amino acid substitution M15/L16 between nonmuscle and muscle actin does not alter the overall topology of the nucleotide-binding site. Instead, the larger side chain of M16 in muscle actins, located in a loop that protrudes into the nucleotide-binding cleft, acts as an extended lid that flanks the active site and shields the phosphate groups of ADP (Figure 2, Figure 2 —figure supplement 1).”

3. Are there compensatory substitutions locally?

The following sentence in the “results” section addresses compensatory substitutions and differences between actin isoform.

“Amino acids M15/L16 and M304/M305 form hydrophobic interactions in the nucleotide- binding cleft active site (Figure 2). Amino acid substitution M15/L16 between nonmuscle and muscle actin does not alter the overall topology of the nucleotide-binding site. Instead, the larger side chain of M16 in muscle actins, located in a loop that protrudes into the nucleotide-binding cleft, acts as an extended lid that flanks the active site and shields the phosphate groups of ADP (Figure 2, Figure 2 —figure supplement 1). Near the nucleotide- binding cleft active site, amino acids C10 and V17 in muscle actins are substituted with V9/I9 and C16 in β/γ-actin (Figure 3, Figure 1—figure supplement 1C). These reciprocal amino acid substitutions maintain the overall oxidation-reduction environment within filamentous actin isoforms which is important for its dynamic properties and the interaction with some regulatory proteins (51-54)”.

3. Amino acid substitutions: Figure 3 provides an interesting overview, but knowing the distributions of these substitutions is really only useful with a deeper analysis at the level of side chains, as noted in (3). At the overview level the absence of substitutions in SD2 is the most interesting observation. Are the residues in SD1 and SD3 that interact with the D-loop also conserved? Posing and answering this type of deeper question would make this paper much stronger.

We thank the reviewer for this suggestion. We analyzed the longitudinal and transverse interprotomer interfaces in all actin isoforms at the side chain level (new *Figures 4 and 5*) to provide a deeper structural analysis of our four strictures of actin isoforms. We also analyzed D- loop interactions in all actin isoforms and found that they are conserved. The following statements have been included in the “results” and the “discussion” of the revised manuscript.

“Next, we analyzed longitudinal and transverse interactions at interprotomer interfaces in our four structures of actin isoforms (*Figure 4, Figure 5, Figure 4 —figure supplement 1*). We found that longitudinal interactions are mainly mediated by hydrophobic amino acids that are likely to enable interactions with water molecules that were recently shown to mediate interprotomer contacts within the filament core (*Figure 4, Figure 4 —figure supplement 1*) (55). Amino acid substitutions at the longitudinal interprotomer interface include T200/V201, C271/A272, and V286/I287. These substitutions may influence the stability of the promoters based on their ability to interact with solvent (55). The transverse interprotomer interface is formed through hydrophilic and hydrophobic interactions (*Figure 5, Figure 4 —figure supplement 1*). In contrast to interactions at the longitudinal interface, these interactions are direct and appear not to be mediated by solvent. Notably, no amino acid substitutions between actin isoforms are present at the transverse interprotomer interface. Interactions with the D-loop located in SD2 are conserved in our structures of actin isoforms, underlining the critical and conserved role of the D-loop to mediate interprotomer interactions. At the intersection of the longitudinal and transverse interprotomer interfaces, H39/40 and H172/H173 together with M43/M44 act as central anchors.”

“The comparison of longitudinal and transverse interprotomer interfaces revealed that the transverse interface is highly conserved compared to the longitudinal interprotomer interface that features several amino acid substitutions. Further, our analysis showed the presence of conserved amino acids (H39/H40, H172/H173, M43/M44) at the intersection of the longitudinal and transverse interprotomer interfaces. These residues were previously shown to be highly susceptible to oxidative stress caused by reactive oxygen species (ROS) and suggest a possible implication for the filament stability of actin isoforms (11).”

“Further, the absence of amino acid substitutions in SD2 and conserved interactions between the SD2 D-loop and amino acids of SD1 and SD3 suggests selective pressure to maintain this critical structural element.”

4. Posttranslational modifications: the text emphasizes that these actin isoforms have their native posttranslational modifications, which are exactly the same ones (acetylated N-terminus and 3-methyl histidine) present in the actin subunits used for previous filament structures, so this is not novel.

We agree with the reviewer that, given the strong conservation of Nt-acetylation and H72/73 methylation in vertebrate actins, these modifications are expected to be present in previous structures of filamentous actin. Indeed, the methylated H73 has been resolved in a recent cryo- EM structure of filamentous skeletal muscle α-actin (Chou and Pollard, Proc Natl Acad Sci USA, 2019). However, to our best knowledge, the acetylated N-terminus has not been resolved in previous structures of filamentous actins due to the lack of observable density for parts of the N- terminus region. We could resolve the entire N-terminus region including Nt-acetylation for β- and γ-actin and the N-terminus region starting from amino acids 2 and 4 for cardiac muscle and skeletal muscle α-actin, respectively. We would also like to highlight that the filamentous structures of bare cardiac muscle α-actin, β-actin, and γ-actin have not been reported in the past and therefore, our structures will complement the repertoire of available structures of filamentous actin isoforms bound to interacting partners. The following sentence has been included in the “discussion” section of the revised manuscript.

“Prevalent PTMs of actin isoforms include acetylation and methylation. Nt-acetylation and H72/H73 methylation are highly conserved in vertebrate actins and therefore expected to be present in most if not all previous structures of filamentous F-actin. However, the methylated H72/H73 has been only recently resolved in the structure of skeletal muscle α-actin (27). The high resolution of our structures allowed us to resolve the methylated H72/H73 in all our structures. Further, we could resolve the Nt-acetylated N-termini in filamentous β-actin and γ-actin.”

5. The section "Myosin modulates actin filament structure" is largely a review of the literature that does not depend heavily on the new structures. It might be left for a review rather than being in a primary research paper.

We respectfully disagree with the reviewer. Based on our four experimental structures of actin isoforms, we revealed that the N-termini of muscle and nonmuscle actin isoforms have different conformations that are likely to contribute to distinct binding interfaces for myosin motors and ABPs. This observation is critical because it provides a structural explanation for the reported differences in the extent actin isoforms can activate the kinetic and functional properties of myosin motors (Muller et al., PLoS One, 2013) which likely translates into the physiological capacities and limitations of the motors in cells. Based on the suggestions of the other reviewers, this section has been expanded and better discussed in the context of prior research in the field in the revised manuscript.

6. Clinical mutations: the text states "our structures enabled us to reveal the location of disease-causing mutations." In fact, the new structures were not necessary to locate these sites, which were known from previous structures or could be inferred from homology models. Therefore, the section on "Amino acid substitutions overlap with the location of mutations and PTMs in actin isoforms" is largely a literature review that does not depend on the new structures. Figure 6 with very low magnification renderings is not useful without consideration of the details at the side chain level.

We thank the reviewer for the comment. The sections focusing on PTMs and the analysis of actin mutations have been deleted.

Suggestions and comments about the text and figures:I would be better to write "skeletal muscle α-actin" throughout.

We have changed instances of “skeletal α-actin” to “skeletal muscle α-actin” in the revised manuscript. We also changed instances of “cardiac/smooth α-actin” and “smooth γ-actin” to “cardiac/smooth muscle α-actin” and “smooth muscle γ-actin” for consistency.

The text states "The gene products (for actin isoforms) are structurally and functionally conserved among eukaryotes.…" Do the authors mean "among vertebrates?" Surely not among all eukaryotes.

We thank the reviewer for this comment and apologize for our oversight. We have replaced the “eukaryotes” with “vertebrates” in this sentence.

"Differential" in the following a poor choice of words: "isoforms display differential biochemistries.."

We thank the reviewer for this comment. The word “differential” has been changed to “different”. The sentence now reads: “Actin isoforms display different biochemistries, cellular localization, and interactions with myosin motors and actin-binding proteins (ABPs).”

"We show that the defining feature used to regulate the interaction with binding proteins is the divergent N-terminus…" but the full N-termini are resolved in only two of the structures.

We would like to emphasize that the N-terminus region encompasses the first 9/10 amino acids of mature actin isoforms (Perrin and Ervasti, 2010). While the reviewer is correct that we could only resolve the entire N-terminus region in β- and γ-actin, we could resolve the N-termini of cardiac muscle α-actin and skeletal muscle α-actin starting from amino acids 2 and 4, respectively. This allowed us to determine the orientation of the N-terminus region with respect to the filament shaft with high confidence for muscle α-actins. The comparative structural analysis revealed different orientations of the N-termini of muscle and non-muscle actins that suggest that these structural alterations are likely to define the functional specialization of actin isoforms.

We have included the following statements in the “results” of the revised manuscript:

“The quality of our density maps allowed us to resolve both key PTMs. Specifically, the presence of resolvable density allowed us to model the entire N-terminus including the Nt-acetylated D1 (D1^AC^) and the Nt-acetylated E1 (E1^AC^) in the cryo-EM reconstructions of β- and γ-actin (*Figure 1E*). Like in previous cryo-EM structures of muscle α-actins, there are no resolvable densities for the very N-terminus, including Nt-acetylation, in our density maps of skeletal muscle α-actin and cardiac muscle α-actin (*Figure 1E*), possibly due to non-uniform PTM patterns of native, tissue- purified muscle actins.”

Page 3, lines 25-26. Symbol font missing three times.

We thank the reviewer for catching these typographical errors. It appears that some Greek characters were inadvertently converted into text. We have corrected all instances in the revised manuscript.

Most investigators use nucleotide-binding cleft and barbed end groove rather than "inner and outer cleft."

We thank the reviewer for this suggestion. We changed instances of “inner cleft” and “outer cleft” to “nucleotide-binding cleft” and “barbed end groove” as suggested. We also included respective labels in panel A of *Figure 1 – —figure supplement 1*.

Many of the fonts in Figure 1, supplement 2 are far too small to read.

We increased the font size in *Figure 1 —figure supplement 2* for better readability. Thank you for the suggestion.

Page 7: the text states "Overall, the amino acid sequence divergence is higher between muscle and nonmuscle actins than within muscle or nonmuscle actin sequences (Figure 1—figure supplement 1C)." This has been known for decades and is not new to this study.

This sentence was not intended to suggest novelty and we apologize if it unintentionally did. The reviewer is correct that this has been known for decades and it is also evident from the sequence alignment shown in *Figure 1 – —figure supplement 1*. In response to a critique raised by reviewer #1, we have re-worded the sentence for clarity. It now reads: “Overall, the amino acid sequence is more conserved among muscle actins than between muscle and nonmuscle actins.” We believe that the sentence is valuable for the non-expert reader and that it aligns with results from our comparative structural analysis that shows different orientations of the N-termini of muscle and nonmuscle actins that are likely to define isoform-specific interfaces with myosin motors and ABPs.

Page 10-11: "We found that the binding of myosin to actin causes subtle conformational changes in the actin filament (Figure 4)." These conformational changes should be described and interpreted in more detail, but the whole section should probably be deleted.

We thank the reviewer for this comment. In response to this reviewer and reviewer #1, we expanded the respective section in the manuscript and included a comparison with the structure of cardiac myosin-2 bound to the native thin filament. We also described structural differences that we observed in more detail. Further, two videos (*Figure 6 – video 1, Figure 7 – video 1*) have been included to better visualize conformational changes in SD2. The following sentences have been included in the “results” of the revised manuscript.

“Structural comparison of our structures of bare actin isoforms and previous structures of myosin- bound actins shows that SD2 adopts a different conformation (*Figure 6A, B, Figure 6 —figure supplement 1, Figure 6 – video 1*). A direct comparison of bare γ-actin with γ-actin/NM2C complex shows the subtle inward movement of the D-loop with an RMSD of ~1.25Å in the myosin-bound state, suggesting that myosin-binding induces subtle changes in the barbed end groove that do not change the separation between SD1/SD3 and SD2/4. Notably, the inward movement of the D-loop (RMSD ~ 0.91 Å) is also observed in the recent structure of cardiac myosin bound to the thin filament compared to our structure of bare cardiac muscle α-actin (*Figure 6 —figure supplement 1*), suggesting that myosin and not the regulatory proteins tropomyosin and troponin drive this structural change.”

Page 12: How does the "subtle inward movement of the D-loop" impact the interactions along the long pitch helix? Again, did "myosin-binding induces subtle changes in the outer actin cleft" change the separation between SD1/3 or SD2/4?

In response to the question raised by this reviewer, we have included two new videos (*Figure 6 – video 1, Figure 7 – video 1*) that show conformational changes in SD2 in bare versus myosin- bound actin and included the following statement in the “results” section of the revised manuscript.

“The conformation of the D-loop, located in SD2, differs with an RMSD of ~0.95-1.25Å (*Figure 6A, B*). A direct comparison of bare β-actin with γ-actin/NM2C complex shows the subtle inward movement of the D-loop with an RMSD of ~1.25 Å in the myosin-bound state, suggesting that myosin-binding induces subtle changes in the barbed end groove that do not change the separation between SD1/SD3 and SD2/4. Notably, the inward movement of the D-loop (RMSD ~ 0.91 Å) is also observed in the recent structure of cardiac myosin bound to the thin filament compared to our structure of bare cardiac muscle α-actin (*Figure 6 —figure supplement 1*), suggesting that myosin and not the regulatory proteins tropomyosin and troponin drive this structural change.”

Reviewer #3:In this study, the authors solve high-resolution structures of bare Mg^2+^.ADP homotypic filamentous actin using pure skeletal α-, cardiac α-, cytoplasmic β- and cytoplasmic γ-actin, including their N-termini, which have been previously elusive to structural studies. They use these data to analyze the differences in protein binding and PTMs, which likely contribute to divergent actin isoform functions in vivo.The major strengths of the methods and results is the use of purified actin isoforms, some of which, e.g., γ cytoplasmic actin, have been previously impossible to obtain biochemically. Furthermore, this is the first resolution of the N-termini of actin protomers, which have been excluded from all the previously published structures. This is important because the N-termini harbor the major divergence between actins and thus can potentially offer key structural/functional insights into their differences in vivo.By comparing the structures of previously published myosin-actin complexes with the bare actin structures, the authors identify subtle differences between the myosin bound and bare actin filaments, with some isoform specific differences dependent on both the myosin and the actin involved in the interaction. The authors also provide a detailed summary of where disease causing mutations in actin isoforms map on the actin structure and suggest how modifications of the physiological "actin-PTM code" could be the molecular basis of diseases caused by mutations in actin proteins.Method-wise, the authors have achieved their goal of solving actin isoform structures. However, the novelty of the conclusions is somewhat overstated and would need to be presented more in the context of prior work in the field, which predicted the N-termini to be important in maintaining actin isoforms' divergence and their role in myosin binding, as well as some of the implications of the disease-causing mutations and PTMs.Overall, this work will inform the field about the most likely functional determinants that drive the diversity of actin isoforms and "actin-PTM code" in vivo. The results of this study will enable future research elucidating specific actins' functions in different tissues and cell types.

We thank the reviewer for the valuable comments and the positive feedback on our work. We have substantially modified the text to put our observations in the context of previous work in the field.

In my view, experiments and data analysis are sufficient, so all the suggestions below concern changes in the text.Overall changes:1. It would be helpful if the authors included a more prominent discussion on the background on the actin isoforms, including the fact that the amino acid sequences of all these actins are conserved in higher vertebrates, so that there are no heterogeneities introduced into the present study by using actins from different species. It also appears important to include an upfont clarification about the source of the actin isoforms used in this study, including the fact that some of them are native and some expressed in Pichia pastoris. While these details can be gathered from the manuscript overall, I felt that stating them more clearly at the beginning could help orient the reader.

We thank the reviewer for this valuable suggestion. We have expanded the “introduction” which now includes more background on actin isoform expression as it relates to the mixing and matching with myosin motors. The following sentences have been added to the revised manuscript:

“Most cells maintain a defined ratio of actin isoforms with muscle and nonmuscle actins representing the main isoforms in muscle and non-muscle cells, respectively (3-8). Actin isoforms have specific and redundant roles in cells and display different biochemistries, cellular localization, and interactions with myosin motors and actin-binding proteins (ABPs) (2, 3, 9-19).”

In addition, we have included the following statement in the “introduction” to point the reader towards the source of the proteins used for structural studies and the conservation across vertebrates

“Specifically, we used our previously established Pick-ya actin method to recombinantly produce human β-actin and γ-actin in an engineered *Pichia pastoris* strain that expresses the human N- acetyl transferase NAA80 and histidine methyl transferase SETD3 to ensure uniform Nt- acetylation and methylation of H72/H73, a conserved PTM profile of vertebrate actins (42). Skeletal muscle α-actin and cardiac muscle α-actin were purified from rabbit skeletal muscle and the left ventricle of a bovine heart, respectively. At the protein level, all actin isoforms are conserved across vertebrates, allowing us to compare our structures to previous structures of filamentous actin from other vertebrate species in the correct physiological context (*Figure 1 —figure supplement 1*) (2, 9, 22).”

2. While the novelty of analyzing multiple actin isoforms and resolving the actins' N-termini is without question, some of the other novelty claims in this paper are overstated. Other studies mapped actin disease-causing mutations on the structure. Even more importantly, it has been long known that N-terminus is the major source of differences between actin isoforms, based on sequence alignments. The role of the N-termini in the interactions with actin-binding proteins and particularly myosin has also been proposed. The authors should word their findings in the context of these prior studies.

We thank the reviewer for this comment. In response to the editor’s summary, the section on disease-causing mutations has been removed.

Actin is a highly conserved molecule and has been subject to several hundreds of research studies and the importance of the N-terminus in making interactions with myosins and actin- binding proteins has been well-established in the field. While there are many elegant studies (Muller, PLoS One, 2013; Abe *et al.*, Biochem Biophys Res Commun, 2000, Cook *et al.*, J Biol Chem, 1992, Chin *et al.*, J Biol Chem, 2022), including work from our groups, that show the importance of the N-terminus in comparative biochemical, functional, and cell biological studies, there is, to our best knowledge, no study that reported different conformations of the N-termini in filamentous actin isoforms that are likely to form the structural basis of the reported differences/characteristics. While alignments are essential tools in comparative sequence analysis of proteins of interest, they rely on the sequence and/or structural information (Carpentier and Chomilier, Bioinformatics, 2019). This makes it very unlikely that the different positions of the N-termini between bare muscle and nonmuscle actins would have been predicted solely based on a sequence alignment or structure-based alignment, as the structures of bare cardiac muscle α-actin, β-actin, and γ-actin were previously unknown. Thus, our experimental structures of actin isoforms allowed us to identify the key structural signatures that were unlikely to be predicted based on a sequence alignment alone and will provide the structural basis for future studies on actin isoforms. As suggested by the reviewer, we have revised our manuscript and to better word our findings in the context of prior studies.

3. Stating that the authors analyzed the effect of Nt-acetylation of non-muscle actins on the interaction with myosin motors (Page 12, Line 12) is not quite accurate. Based on the structures of Nt-acetylated actins that the authors solved, they can speculate that the N-terminal acetyl group may contribute to the myosin loop 2 interaction with actin, but they have not "analyzed" the effect of Nt-acetylation formally. Furthermore, the actin N-terminus is highly negatively charged even without the Nt-acetylation. The authors propose that the additional negative charge introduced by the acetyl group likely enhances the long-range electrostatic interactions with the positively charged myosin residues in loop 2. This is speculative and does not appear to be strongly supported by data. This should be reflected in the text.

We apologize for the misleading wording. We modified the sentence and changed the word “analyzed” to “compared”. It now reads:

“Next, we compared the position of the exposed actin N-terminus on the interaction with myosin motors”.

Previous research has shown that Nt-acetylation contributes to normal actin filament structure, facilitates filament nucleation, and modulates the interaction with other protein partners including myosin motors (Abe *et al.*, Biochem Biophys Res Commun, 2000, Cook *et al.*, J Biol Chem, 1992, Chin *et al.*, J Biol Chem, 2022). For myosins, it has been shown that actin Nt-acetylation results in a ~4-times smaller Kapp in steady-state ATPase assays compared to non-acetylated actin without changing the kcat (Abe *et al.*, Biochem Biophys Res Commun, 2000), suggesting that it strengthens weak electrostatic interactions between both proteins.

The following statement has been included in the “discussion” of the revised manuscript:

“For example, previous in vitro experiments with recombinant proteins revealed isoform-specific differences in the interaction between muscle and nonmuscle actin isoforms and individual myosin motors (16). Notably, predominantly nonsarcomeric myosins have a faster ATPase activity and in vitro sliding velocity when assayed with nonmuscle actins compared to muscle actin, suggesting selective fine-tuning of the functional competencies of myosin-actin combinations with implications for cell function (16). These in vitro observations are supported by our structural data that suggest that the different orientations of the N-termini of actin isoforms contribute to the formation of different binding interfaces with myosin motors (Figure 7). Previous work also revealed differences in the extent to which the highly conserved β-actin and γ-actin isoforms (*Figure 1—figure supplement 1*) can activate the kinetic and functional activity of the same motor in vitro (16). Based on our structural and previous biochemical studies, we speculate that the biochemical properties of the very N-terminus of actin isoforms are key contributors to the formation of isoform-specific interfaces with myosin motors (16, 56, 71). For instance, the very N- termini of mature β-actin (DDD) and γ-actin (EEE) have different pKa values, resulting in a higher negative electron charge density in β-actin compared to γ-actin that is further increased by Nt- acetylation in both isoforms. These differences are likely to contribute to distinct interactions with the positively charged loop-2 and other actin-binding elements in the motor domain of myosin family members during the formation of the actomyosin interface.”

Specific line comments:1. Page 3, Line 25-26: g-actin should be "γ-actin". Also, skeletal a- and cardiac a-actin should be skeletal "α- and cardiac α-actin".

We thank the reviewer for catching these typographical errors. It appears that some Greek characters were inadvertently converted into text. We have corrected all instances in the revised manuscript.

2. Page 3, Lines 27-29: The authors refer to skeletal muscle actin N-terminus being unresolved and refer to Figure 1B. However, Figure 1B is labelled as cardiac actin. Also, looking at the density of the N-termini in Figures 1A-D, Figure 1B a-1 protomer seems to have the least density at what would be the N-terminus, which would make this the skeletal muscle actin? Is this a labeling error? Or is this because Figure 1B appears to be slightly rotated around the filament long axis compared to the other three structures, thus not showing the N-terminus fully at this angle in a-1 protomer. This should be corrected.

We apologize for this oversight and the labeling error. We meant to refer to *Figure 1E* and corrected this error in the revised manuscript. We now also include videos of actin isoform protomers to show the quality of densities and our reconstructions (*Figure 1 – videos 1-4*) for better visualization.

3. Page 8, Line 26: Actually, V9 is in β-actin, while in γ-actin it is I9. While this is a conservative substitution, it would be interesting to know if this has any effect on the nucleotide binding and/or dynamic properties, or interaction with some regulatory proteins. The authors should comment on this, if possible.

We thank the reviewer for catching this mistake. We have corrected it in the revised manuscript. We have performed an extensive literature search but the role of V9/I9 has, to our best knowledge, not been experimentally addressed. Based on biochemical studies with recombinant human β- and γ-actin (20-25% contaminated with endogenous insect cell actin), Bergeron *et al.*, speculate that observed differences in polymerization kinetics between both isoforms can be attributed to the V9/I9 substitution based on their location in the core of secondary structural elements that form part of the nucleotide-binding cleft (Bergeron *et al.*, J Biol Chem, 2010). However, a possible role for the three N-terminal amino acid substitutions could not be excluded (Bergeron *et al.*, J Biol Chem, 2010).

We analyzed the position of V9/I9 in our structures. We found that this residue is part of a hydrophobic pocket in the core of SD1 that stabilizes part of the nucleotide-binding cleft. Since it is not surface exposed, it is unlikely to directly interact with actin regulatory proteins.

4. Figure 4: The authors refer to the rigor state as apo in the text. Either the rigor nomenclature or the apo nomenclature should be followed for consistency.

We have changed all instances of “apo” to “rigor” in the revised manuscript. We thank the reviewer for catching this oversight.

5. Page 12, Line 9: I believe the authors are referring to Figure 4B, not Figure 5B.

We thank the reviewer for catching this error. It has been fixed in the revised manuscript.

6. Page 12, Line 15 and Figure 5: The authors refer to Figure 5B, but I believe they are actually talking about Figure 5A. Figure 5A labeling needs to be clearer. I believe the teal and yellow labels are referring to Myosin-1B bound α-actin, and the magenta to NM2C bound γ-actin, while the blue and dark red are bare cytoplasmic actins. Instead of labeling the structures with just the motor protein, the authors need to label which actin isoform is bound to the labeled motor. It might help if the other colors in this figure are suffused, so that the ones emphasized stand out more.

We thank the reviewer for catching this error. It has been fixed in the revised manuscript. We also revised the labeling in this figure (now *Figure 7*) for better readability.

7. Page 12, Line 26: The authors talk about the shorter loop-2 in M1B being less efficient at stabilizing the actin N-terminus. It is unclear if they are referring to how the α-actin N-terminus points in a different direction in M1B bound structures compared to NM2C bound γ-actin N-terminus. If so, why is this referred to as stabilizing the N-terminus?

We thank the reviewer for this comment and apologize for the ill-worded sentence. M1B has a short loop-2 compared to other members of the myosin superfamily that, together with a restricted loop-2 geometry, limits its ability to contact the actin surface (Mentes *et al.*, Proc Natl Acad Sci USA, 2018). By comparing the previous structures of rigor and Mg^2+^.ADP M1B bound to skeletal muscle α-actin and the structure of NM2C bound γ-actin with our structures of bare β-actin and γ-actin, we noticed that the short loop-2 of M1B does not interact with the skeletal muscle α-actin N-terminus while the longer loop-2 of NM2C (and likely other myosins) interacts with the N- terminus of γ-actin. We have revised the sentence as follows:

“Our structural comparison also revealed that a shorter loop-2, as it is found for example in M1B, is less efficient in stabilizing the actin N-terminus due to geometric constraints that limit its ability to interact with and pull the N-terminus closer to the filament surface in the strong binding states compared to a longer loop-2 as it is found in NM2C (*Figure 7A*).”

[Editors' note: further revisions were suggested prior to acceptance, as described below.]

Although both manuscript text and figures have been modified in the revised version, there are still several issues that should be addressed before the manuscript is acceptable for publication. Reviewer #2 has, therefore, provided a detailed list of revisions (see below). Thus, both the manuscript text and several figures should be extensively revised to improve the data presentation, as well as the focus of the manuscript.Reviewer #2:The new structures seem to be well done and are a valuable contribution. The authors made many changes in response to the first round of reviews. Unfortunately, these changes did not improve the paper substantially. The authors revised the figures, but the new figures have the same problems as the originals. The text of the article has much new material (in response to the reviews) but the result is that the presentation is now poorly organized, and still contains unwarranted or inaccurate statements.Making acceptable figures and rewriting the text should be straightforward but so far has escaped the authors. I would like to help but cannot rewrite the paper. The following is an incomplete list of issues that need to be addressed:• Title: Not appropriate. The paper mostly describes some new structures. The only mechanistic advance is the first views of the complete N-termini of two actins with some ideas about how they might influence interactions with myosin.

We thank the reviewer for the suggestion. We changed the title to “Structural Insights into Actin Isoforms”

• Abstract: The focus is wrong; it contains nothing specific about how the new structures are the same or different from each other. What is a "retropropagated structural change?"

We have removed the sentence “Retropropagated structural changes further show that myosin binding modulates actin filament structure” from the abstract. Further, we have deleted the expression/term “retropropagated structural change” from the “discussion” section of the revised manuscript.

• Introduction: The text attributes to differences in actin isoforms "the formation of diverse cellular actin networks with distinct compositions, architectures, dynamics, and mechanics that enable fundamental cell functions including adhesion, migration, and contractility." Small differences in the isoforms may facilitate the formation of different structures but are unlikely to be causative as claimed. Actin binding and signaling proteins surely are the dominant factors.

We have changed the sentence in response to the critique raised by reviewer. It now reads: “Driven by the dominating action of ABPs and signaling proteins, differences between actin isoforms may facilitate in the formation of diverse cellular actin networks with distinct compositions, architectures, dynamics, and mechanics that enable fundamental cell functions including adhesion, migration, and contractility.”

Results• Maps and models: The text states "Our cryo-EM maps allowed us to build unambiguous models of actin isoforms in which secondary structure information including the side chains, the nucleotide and associated cation (Mg^2+^·ADP), and PTMs were apparent from the densities (Figure 1E, Figure 1 – —figure supplement 2)." The models were not built de novo from the maps. Rather a previous structure was docked into the maps followed by automated refinement procedures. I expect that the backbones changed very little during refinement, but this is not stated. It would be interesting to know how much refinement adjusted the side chains and handled the substitutions.

We thank the reviewer for this comment and would like to respectfully point out that we make no claims in the manuscript about the de novo building of the maps. While the high resolution and quality of our maps of four actin isoforms would have allowed de novo model building, we used a reference structure as a starting model to build our maps, as detailed in the “material and methods” section. It is common practice in the field to generate initial models based on a reference structure, if available. We manually built numerous parts in our structures of four actin isoforms including the D-loop, N- and C-terminal residues, and amino acid substitution between isoforms. Further, we modeled PTMs and placed the Mg^2+^.ADP in the nucleotide-binding cleft active site. For example, manual model building revealed different rotamer positions in the nucleotide-binding cleft active site of actin isoforms (*Figure 1*, *Figure 2*, *Figure 2 —figure supplement 1*). All models were subjected to several rounds of iterative model building – manual and real space refinement – based on our high-resolution maps in Coot and Phenix to build unambiguous models of four actin isoforms, as mentioned in the “material and methods” section.

The figures do not allow a reader to evaluate the quality of the maps and models, since the maps are shown at very low contour levels, into which almost any side chain would fit. Furthermore, some of the new surface representations of the maps (Figure 1E, Figure 1S1B, Figure 2) are shown with low contrast and almost no depth cues making it difficult to see how the models fit into the maps even in the videos. Other renderings (Figure 1S3) are too dark to see details.

The scientifically accurate visualization of 3D protein maps, structures, and atomic interaction in 2D figures to everyone’s satisfaction is a challenge and highly subjective. For example, the original meshwork representation of the maps in *Figure 1E* and *Figure 2* has been replaced with a transparent surface representation in response to critiques raised by reviewer #1 during the first revision of our manuscript. As per the reviewer’s suggestion, we have updated *Figure 1 —figure supplement 3* in the revised manuscript. Thank you for this suggestion.

The models are rendered by an unspecified method with a continuous backbone that differs in thickness between figures and seems to be some kind of smoothened average rather than representing the atoms and bonds. The stick figures of side chains are better. Unfortunately, without stronger depth cues or thinner Z-slices, most of the details are lost in a confusing maze of overlapping details. To make matters worse, some points of view are not helpful: for example, viewing a histidine ring on edge is the worst way to see a methyl group. Similarly, the views of the active site are not as clear is previous papers from other labs.

We thank the reviewer for this comment. We used the licorice representation in ChimeraX to show the backbone. We now include this information in the figure legends, as applicable. Our original goal was to show the position of the methylated H72/73 in actin isoforms with respect to the nucleotide in *Figure 1 —figure supplement 3*. Based on the critique raised by the reviewer, we have updated *Figure 1 —figure supplement 3* to show the methylated H72/73 in a different orientation for all four actin isoforms.

The four new videos of the maps, backbone models and side chain models offer some help, but they have problems with low contrast maps and overlapping structures as noted above. Even at half speed the images rotate too quickly to see details such as the N-termini.

We thank the reviewer for this comment. We have slowed down the speed of the *Figure 1 – videos 1- 4* for better visibility. We now also include an additional video (*Figure 1 – video 5)* that shows the reconstruction of the N-termini of actin isoforms including the densities in both surface and mesh representation. All electron densities are contoured atß 1.8 to 2α. This information has been included in the respective figure legends.

The deep colors used in Figure 1A-D do not show the subunits clearly. A color tinted version of Figure 1 – supplement 1 would be much better.

As suggested by the reviewer, the deep colors have been changed to light colors in *Figure 1A-D* in the revised manuscript. We also included a color-tinted version of panel A in *Figure 1 —figure supplement 1*.

The font size is too small in Figure 1S2: Here are the standards for font size in figures (from JBC): Use a sans serif font such as Arial or Helvetica. Use regular font style, not bold. Use appropriately sized numbers, letters, and symbols so they are no smaller than 2 mm after reduction to a 1, 1.5 or 2 column width. Superscript and subscript characters are not excluded from this rule. Numbers, letters and symbols used in multi-paneled figures must be consistent, that is all the same font and size.

Thank you for this suggestion. We followed the *eLife* author guide to prepare this manuscript. As per the suggestion of the reviewer, we have increased the font size of the labels of the local resolution maps shown in *Figure 1 —figure supplement 2*.

• Comparison of the structures. The organization of this section needs to be rethought. Here is a possible outline:1. Comparison of helical parameters and subunit backbones.Lines 130-131 belong at the beginning of the structure comparisons "No significant changes (this should be differences, since the isoforms do not interchange) in the pitch of the actin helix were observed between our structures of actin isoforms, emphasizing their overall conserved filamentous structure in the absence of myosin motors or ABPs."

Thank you for this suggestion. We have moved the sentence to the beginning of the section “Similarities and differences between actin isoforms”. Further, we have changed the word “changes” to “differences” in the respective sentence and throughout the manuscript, as appropriate.

Rather than being tacked on as an afterthought at the end of a paragraph about the maps, the following should be the topic sentence for a paragraph in this section: "The superposition of all actin isoform structures shows a root-mean-square deviation (RMSD) between 0.83Å to 1.04Å, indicating an overall similar topology." Are the RMSD's for all atoms or more appropriately the backbones? I expect that within experimental error the backbones are identical (rather than "overall similar").

We thank the reviewer for the suggestion. The sentence is now the first sentence of the section “Similarities and differences between actin isoforms”. We now also include actin helical parameters (helical rise: ~ 27.6 Å to 28 Å, twist: ~ -166.5° to -168°) for all structures of actin isoforms in the “results” section. The RMSD values are for the backbone atoms, not for all atoms, suggesting that the structures are similar and not identical.

2. Comparison of side chains. Lines 97-113 are common knowledge, so they can be deleted unless used to make specific points of comparison between the structures.

While we agree with the reviewer that these lines describe the overall architecture and features of the actin filament, we believe that they are valuable as an introduction to our work and the readability of our work for the non-expert reader.

Figure 5 showing the locations of the substitutions is excellent. However, identifying these locations (lines 116-127) does not depend on the new structures. Instead, the new structures allow for the first examination the effects of substitutions on local structures (as requested in the first review). Consider are the following questions. Are the side chain rotomers identical in the isoforms or do they differ? Do the orientations of the side chains differ where there are substitutions? As requested in the first review (but not provided), are there compensatory substitutions at other positions when alternative side chains have different volumes?

We thank the reviewer for the positive feedback. We now include two new figure panels (*Figure 4 —figure supplement 1C,D*) that show the conformation of substituted amino acid side chains in the transverse and longitudinal interprotomer interface. We intentionally only show substituted amino acid side chains – not all side chains – of skeletal muscle α-actin and ß-actin as representatives for muscle and nonmuscle actin isoforms to avoid crowding. The new panels show different orientations for some side chains as well as their specific interfaces. Compensatory substitutions for example include L175/M176 and C271/A272. The reciprocal amino substitution is likely to contribute to the maintenance of the oxidation-reduction environment as do other reciprocal substitutions near the nucleotide-binding cleft active site as described in the manuscript.

Line 155: It is not true that "side chain of L16 in muscle actins" is larger than methionine. Both have 4 heavy atoms.

We apologize for the oversight. We meant “longer”, not “larger”. Please see Author response image 1. We changed the sentence to: “Instead, the longer side chain of M16 in muscle actins, located in a loop that protrudes into the nucleotide-binding cleft, acts as an extended lid that flanks the active site and shields the phosphate groups of ADP”.

**Author response image 1. sa2fig1:** 

3. Comparison of subunit interactions in filaments.Lines 163-177: The key findings are buried in the paragraph. (Finding 1) The transverse interprotomer interfaces (also called the short pitch helix interfaces) are identical in the actin isoforms (cite Figure 4 as the evidence and make it clear in the legend that these interfaces are identical). (Finding 2) Interactions of the D-loop in SD2 with SD1 and SD4 of the neighboring subunit are identical in the actin isoforms. If (2) is correct, how does it relate the statement "Amino acid substitutions at the longitudinal interprotomer interface include T200/V201, C271/A272, and V286/I287." Are these strictly along the long-pitch helix? What are the partner residues contacted in the adjacent subunit? Where are these differences and which interactions are affected? The rest of the material in this paragraph was already known and does not need to be repeated. Note that this study did not reveal "that longitudinal interactions are mainly mediated by hydrophobic amino acids."

We re-analyzed our structures and located a single amino acid substitution (V286/I287) in the transverse interprotomer interface. We show and highlight this amino acid substitution in *Figure 3 —figure supplement 1*, *Figure 4 —figure supplement 1C,D*, and *Figure 5*. We now also included two new panels (C, D) in *Figure 4 —figure supplement 1 that show* select amino acid substitutions in the longitudinal and transverse interprotomer interface, respectively. Amino acid substitutions T200/V201, C271/A272, L175/M176, Q224/N225, and F278/Y279 are much closer to the longitudinal interprotomer interface than to the transverse interprotomer interface. In contrast, amino acid substitution V286/I287 is located at the junction of the longitudinal and the transverse interprotomer interface (*Figure 3 —figure supplement 1, Figure 4 —figure supplement 1*).

We included the following sentence in the “results” section of the revised manuscript: “Amino acid I287 is in hydrophobic contact with I208 and L242 in muscle actins and buries a surface area of ~113Å^2^, whereas the interaction between V286, I207, and L241 in nonmuscle actins is less hydrophobic and buries a surface area of ~83Å^2^ in the transverse interprotomer interface (*Figure 4 —figure supplement 1)”*.

We are surprised by the statement that “Interactions of the D-loop in SD2 with SD1 and SD4 of the neighboring subunit are identical in the actin isoforms” as we do not describe interactions between the D-loop and SD4 in the manuscript. In our manuscript we wrote “interactions with the D-loop located in SD2 are conserved in our structures of actin isoforms, underlining the critical and conserved role of the D-loop to mediate interprotomer interactions”. We also wrote “the absence of amino acid substitutions in SD2 and conserved interactions between the SD2 D-loop and amino acids of SD1 and SD3 suggests selective pressure to maintain this critical structural element”.

I do not understand what Figure 3 supplement 1 or Figure 4 supplement 1 add to the paper if these interfaces are identical in the various isoforms.

*Figure 3 —figure supplement 1* is an overview that shows the location of individual and superimposed amino acid substitutions in actin isoforms with respect to the longitudinal and transverse interprotomer interfaces. We have included labels for key amino acid substitutions in the revised manuscript for better readability. As mentioned above, we re-analyzed the interprotomer interfaces and located a single amino acid substitution in the transverse interprotomer interface. That means that, contrary to our previous statement, the respective interfaces are not identical. We apologize for this oversight and changed the text accordingly.

*Figure 4 —figure supplement 1A and 1B* show the axis that we have used in *Figure 4* and *Figure 5* to analyze interprotomer interfaces. Therefore, it serves as a reference for the orientation of the protomers and the color coding of individual protomers shown in *Figure 4* and *Figure 5*. In addition, we have added two new panels to *Figure 4 – —figure supplement 1* to show only the substituted amino acids along the longitudinal and transverse interprotomer interfaces, respectively.

I do not understand the legend of Figure 5. This looks like the longitudinal interface not the transverse interface, which is shown in Figure 4. In any case the legend should make clear how these subunits are oriented.

Figure 5 shows the transverse interface. We have included a statement that the protomers are oriented according to *Figure 4 —figure supplement 1A, B* in the legends of *Figure 4* and *Figure 5*.

I do not understand the legend of Figure 6, which says this shows "The actomyosin interface of the native cardiac thin filament." If myosin is bound, it is not a native thin filament.

The word “native” was used to indicate that tissue purified thin filaments, not synthetic thin filaments, were used in the work by Risi *et al.* However, we have removed the word “native” from the figure legend and the corresponding text in the revised manuscript to avoid confusion. Thank you for the suggestion.

• Posttranslational modifications.Lines 179-188 are not results from this study and can be deleted.

The paragraph serves as an introduction and as a rationale for why we are interested in resolving the PTMs in our structures. Further, it highlights the power of our Pick-ya actin method to recombinantly produce mature nonmuscle actin isoforms with homogeneous PTM patterns. The paragraph has been updated to make it more concise.

Lines 189-205 are not really about PTMs, rather they are about observing density for previously unobserved residues at the N-termini. I do not agree with the statement "quality of our density maps allowed us to resolve both key PTMs." The maps shown in Figure 1 – supplement 3 are not good enough to identify the methyl group on histidine. As noted above, the ms does not include a high magnification illustration of the maps and models for the four N-termini, so one cannot evaluate whether the acetyl group is present.

We respectfully disagree with the reviewer. Our maps clearly show densities for the Nt-acetylated N- termini of nonmuscle actins and the methylated H72/H72 in all four actin isoform structures presented in this work. We have updated *Figure 1 —figure supplement 3* and added *Figure 1 – video 5* for better visualization of the maps and reconstructions.

• Myosin interactions. This is the most important section of the paper. It will be better with a more concise presentation.Line 223: Start this section with the excellent topic sentence "Myosin binding to actin filaments causes subtle conformational changes in the filament." Following this topic sentence, add lines 224-227, although local changes are more important than the overall RMSDs. Rather than listing the structures on lines 210-222, comment on what you learned from each later in the section. Finish the paragraph with lines 232-243 starting with "SD2 adopts a different conformation…."Continue with "The superimposition of bare… " on line 248.

We thank the reviewer for this suggestion. We have modified this section to make it more concise.

As a general matter, phrases such as "First, we compared," "We found that…," "Next, we compared the position…" and "Our structural comparison also revealed…" are not necessary. Just state the result.

We thank the reviewer for this suggestion. We have removed the phrases in the revised manuscript, as appropriate.

Figure 6 supplement 1 shows the relation of the actin N-terminus to myosin particularly well, although this ribbon diagrams would be better with a wider range of colors for the actins and myosin. This view might be combined in some way with Figure 7 and shown in the main text rather than Figure 6S1. Showing the map densities for these actin N-termini is essential to convince readers that their conformations differ. How does one know if the conformations in bare filaments are the same with myosin bound?

We thank the reviewer for this suggestion. Figure 7 only focuses on the N-terminus of actin (bare vs myosin-bound) and it already contains 5 structures. We added *Figure 6 —figure supplement 1* as per the request of reviewer 1 during the first revision and decided not to combine it with other figures because they get too busy. To avoid repetition and for better visualization, we do not show densities in *Figure 7*. We included videos (*Figure 1 – video 1-4*) to show the densities of bare actins for the entire actin protomers in the first revision as suggested and now include another video (*Figure 1 – video 5*) that shows the N-termini of bare actins with the respective densities of our four structures of actin isoforms. Further, the manuscript contains videos that show the conformation of the actin N- terminus and D-loop (*Figure 6 – video 1*, *Figure 7 – video 1*) in bare and myosin-bound structures.

Line 312: topology is used incorrectly.

We thank the reviewer for catching this mistake. We changed “topological changes” to “structural changes”.

• Discussion. Again, a more concise presentation would be better. You can go straight to the point that the structures of the subunits, filaments and interfaces between subunits in filaments are virtually identical, is spite of modest numbers of amino acid substitutions. The new structures reveal some or all of the N-termini of these actins, which seem to interact differently with myosins bound to the filaments. (Contrary to the conclusion at the end of discussion, the paper has no evidence about how the interactions of the isoforms with ABPs might differ.)

We thank the reviewer for this suggestion. We have modified this section to make it more concise. We have also removed the statement on ABPs from the sentence. It now reads: “In conclusion, we present direct evidence for the structural divergence of actin isoforms that underlies their nuanced interactions with myosin motors”.

Delete lines 281-287 unless the visualization of these PTMs can be made more convincing.

We updated *Figure 1 —figure supplement 3* to provide a better visualization of the methylated histidine in our four reconstructions of actin isoforms as suggested by the reviewer. Further, we included a new video (*Figure 1 – video 5*) that shows the N-termini of all four actin isoforms together with the corresponding densities in surface and mesh representation in the revised manuscript.